# Hidden heatwaves and severe coral bleaching linked to mesoscale eddies and thermocline dynamics

Alex S. J. Wyatt [1] ✉, James J. Leichter[2], Libe Washburn[3,4], Li Kui [3], Peter J. Edmunds [5] & Scott C. Burgess [6]

The severity of marine heatwaves (MHWs) that are increasingly impacting ocean ecosystems, including vulnerable coral reefs, has primarily been assessed using remotely sensed sea-surface temperatures (SSTs), without information relevant to heating across ecosystem depths. Here, using a rare combination of SST, high-resolution in-situ temperatures, and sea level anomalies observed over 15 years near Moorea, French Polynesia, we document subsurface MHWs that have been paradoxical in comparison to SST metrics and associated with unexpected coral bleaching across depths. Variations in the depth range and severity of MHWs was driven by mesoscale (10s to 100s of km) eddies that altered sea levels and thermocline depths and decreased (2007, 2017 and 2019) or increased (2012, 2015, 2016) internal-wave cooling. Pronounced eddy-induced reductions in internal waves during early 2019 contributed to a prolonged subsurface MHW and unexpectedly severe coral bleaching, with subsequent mortality offsetting almost a decade of coral recovery. Variability in mesoscale eddy fields, and thus thermocline depths, is expected to increase with climate change, which, along with strengthening and deepening stratification, could increase the occurrence of subsurface MHWs over ecosystems historically insulated from surface ocean heating by the cooling effects of internal waves.

Marine heatwaves (MHWs) are estimated to have increased by more than 50% globally during the past century, with their frequency and severity likely to continue increasing with ongoing global warming[1–3]. Severe and persistent MHWs have been shown to exert significant impacts on marine ecosystems, leading to species migrations, shifts in community compositions, mass mortality, and biodiversity loss[4–7]. While definitions of MHWs vary, they all aim to statistically describe periods when remotely sensed satellite sea-surface temperatures (SSTs) are anomalously high for sustained periods relative to climatologically derived SST thresholds. The subsurface impacts of MHWs,

which are unlikely to be evident in SST, have rarely been assessed, primarily due to a lack of continuous in-situ temperature observations across the range of depths relevant to coastal ecosystems. The extent to which MHWs extend across depths, the connections between surface heating and subsurface conditions, and thus the impacts on marine organisms and ecosystems, are largely unknown. The vertical structure of MHWs is especially important in determining their impacts on ecosystems encompassing regions of steep vertical temperature gradients[8]. For example, many coral reefs are exposed to the vertically stratified tropical ocean over a depth range that can extend

[1]Department of Ocean Science, The Hong Kong University of Science and Technology, Clear Water Bay, Kowloon, Hong Kong. [2]Scripps Institution of Oceanography, University of California San Diego, La Jolla, CA, USA. [3]Marine Science Institute University of California, Santa Barbara, CA, USA. [4]Department of Geography, University of California, Santa Barbara, CA, USA. [5]Department of Biology, California State University, Northridge, CA, USA. [6]Department of Biological Science, Florida State Universitys, Tallahassee, FL, USA. ✉e-mail: wyatt@ust.hk

from the sea surface to ≥60 m depth where average temperatures may be multiple °C cooler than at the surface.

Coral reef communities on the fore reef slope of Moorea, French Polynesia (17.5 °S, 150 °W; Fig. 1a, b) in the central South Pacific Ocean represent an ideal opportunity to examine the vertical dynamics and impacts of MHWs due to the availability of rare, long-term in situ

oceanographic and ecological data collected concurrently across reef depths from the surface to 40 m since 2005[9]. Here, using a combination of in-situ and satellite observations, we show that the severity of subsurface MHWs and the occurrence of coral bleaching across depths is driven by mesoscale eddy dynamics, which alter thermocline depths and modulate the cooling effect of internal waves.

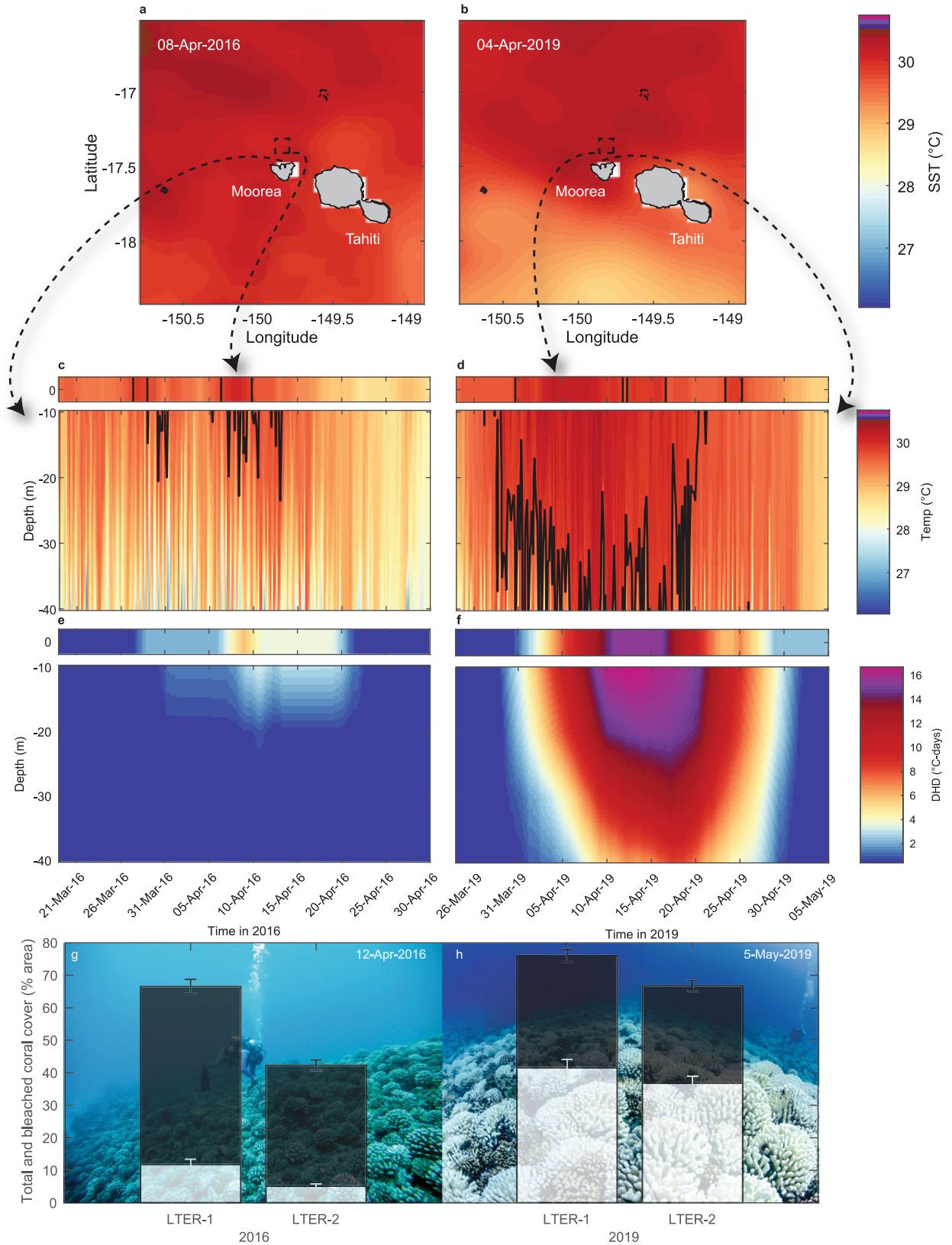

**Fig. 1 | A severe, subsurface marine heatwave (MHW) led to extensive heating and coral bleaching across depths on Moorea's north shore during 2019.** Satellite sea-surface temperatures (SSTs in °C) varied little regionally between **a** 2016 and **b** 2019 MHWs (see Fig. 2 for regional SST patterns), peaking at 30.1 °C on 8 April 2016 compared to 30.2 °C on 4 April 2019, but were locally elevated north of Moorea for longer during 2019 (4 days versus 18 days above the SST-derived bleaching threshold of 29.8 °C in April 2016 and 2019, respectively, within the 0.1° × 0.1° dashed square box shown in **a**, **b**). Subsurface temperatures contrasted markedly during the MHWs, with **c** internal waves reducing subsurface water temperatures during 2016 but **d** high temperatures persisted across depths during

2019. Consequently, **f** a severe and prolonged degree heating day (DHD; °C-days) event occurred across the reef in 2019 (maximum 17 °C-days) compared to **e** weaker heat accumulation restricted to the shallows in 2016 (maximum 3.2 °C-days). Correspondingly, **g** minimal bleaching (white bars, mean ± se) of live coral cover (black bars) was observed during 2016 (here at two of the Long-Term Eco-logical Research (LTER) program sites on the north shore at 10 m depth) but **h** severe, widespread bleaching occurred in 2019 (backgrounds are representative reef-scape images). The black lines in **c**, **d** show the SST-derived bleaching threshold. SST (0 m) values are shown in c-f (top panel) for comparison. The colour scale in **a**, **b** matches that in **c**, **d**. Coastlines in a and b based on Wessel and Smith[77].

## Results and discussion
### Quantifying heating of coastal ecosystems

A variety of metrics have been developed to quantify MHWs, but to date, they have mainly focused on surface heating evident from SST. Surface MHWs have been defined based on exceedance of 90% confidence intervals calculated seasonally from historical SST[10,11]. Defined in this way, the threshold temperature for quantifying a MHW is likely to vary seasonally from, for example, winter to summer. However, the physiological and ecological relevance of seasonally changing thresholds remains unclear, especially for tropical biota such as corals that typically inhabit a relatively narrow range of temperatures close to their physiological thermal limits[12–14]. Assessing heating in the specific context of temperature-induced coral bleaching has instead focused on calculating cumulative degree heating, sensitive to both the magnitude and duration of heating, above a fixed, putative 'bleaching threshold' defined by the local Maximum Monthly Mean SST (MMM); i.e., the mean summer-time peak temperature predicted to initiate coral stress and bleaching. Such heat accumulation has most commonly been expressed as Degree Heating Weeks (DHW in °C-weeks) accumulated over a 12-week period[15,16], or sometimes using monthly SST data to compute Degree Heating Months (DHM in °C-months) over 3 months[17,18]. Numerous studies have documented coral bleaching in shallow water linked to periods of anomalously high SST and accumulated DHW, especially during El Niño events[1,2,4,19–21]. However, DHW generally only explain a limited proportion of the observed variation in bleaching, even for communities in very shallow water (e.g., 50% of bleaching variation at 2 m depth[7]). A number of studies have documented bleaching that was less than that predicted based on contemporaneous SST and DHW. This discrepancy has sometimes been hypothesised to reflect ongoing coral acclimatisation to increasing temperatures and/or shifts in community composition towards more heat-tolerant genotypes and species[22–26]. Bleaching rates higher than predicted by SST, under limited DHW, have also been documented, and can be species-specific and more pronounced in heat-intolerant cryptic species[27]. It is unclear to what extent the relatively limited power of SST metrics such as DHW to predict coral bleaching results from an incomplete description of environmental conditions, a lack of nuance in describing biological thresholds and organisms' reactions to elevated temperatures, genetic variation and cryptic species[27,28], or, most likely, a combination of factors.

Estimates of surface MHW severity are also highly sensitive to both the spatial and temporal scales of SST data considered. Present-day satellite products allow degree heating to be calculated at relatively fine spatial and temporal resolution using SST data measured daily over pixels of ~25 km² size (e.g., NOAA Coral Reef Watch[29]). While many heating assessments in the context of coral bleaching continue to focus on long-term heating based on temperature anomalies accumulated across months (12 weeks or 3 months for DHW or DHM, respectively)[18,20,30], here we focus on higher-resolution heating calculated as Degree Heating Days (DHD in °C-days) using daily SST data. Calculation of DHD over 12-day windows is analogous to the more coarse resolution DHW (i.e., weekly data over 12 weeks)[31] but with different units, finer temporal resolution, and shorter time lags between the actual elevation of environmental temperatures and the

resulting accumulated heating metric (see the DHD and DHW comparison in Fig. S5).

To investigate the role of spatial scale in characterising MHWs, we analysed SST at a range of scales around Moorea between 1985 and 2019: 2° × 2° (~50,000 km²), 1° × 1° (~12,300 km²) and 0.1° × 0.1° (~100 km²) (see Fig. S1). The importance of spatial scales in heating severity apparent at the surface is reinforced by the regional heterogeneity in SST (see Fig. 2 and data animations in the Supplementary Information). Considering heating at local scales and finer temporal resolutions maximises the potential to detect, characterise and compare heating events at the surface in a way that is relevant to in situ MHW conditions (see also Guo et al.[32] for the importance of scales in MHW assessments). For instance, when assessed using NOAA's 'Regional Bleaching Heat Stress Gauges' for the Society Islands, DHW reached 4.54 °C-weeks in April 2016 and 5.35 °C-weeks in May 2019[33], indicating moderate likelihood of bleaching during both events, although inherent to calculations of accumulated DHW, the maximum heating was centred multiple weeks after the in situ heating events actually occurred (e.g., see Fig. S5e, f). Heat accumulation using higher resolution DHD across 12-day windows around Moorea itself (2° × 2°) was more closely aligned temporally to the heating events, suggesting a much higher bleaching risk in 2016 (6.83 °C-days) than 2019 (2.10 °C-days; Table S3). This likely reflects the localised heterogeneity in SST during the 2019 MHW, when hotter surface conditions prevailed north of Moorea (Fig. 1b). At a local scale, assessed using SST within a ~10 × 10 km area north of Moorea (see Fig. S1), heating was similar to regional estimates in 2016 (5.73 °C-days), but in 2019 revealed an intense, localised heatwave over Moorea's north shore (15.4 °C-days; see Fig. 1e, f; Table S1). Because of these demonstrated advantages of higher temporal-resolution analysis, we rely on DHD rather than DHW for our analysis of MHW patterns among years at Moorea.

### Contrasting surface and subsurface heating

Comparing surface and subsurface MHWs is challenging for many coastal ecosystems, even once SST data of appropriately fine spatial and temporal scale are obtained, due to a lack of long-term, in-situ temperature data through which to assess mean climatological patterns below the sea surface. Moorea represents one of the few coral reef systems with consistent in-situ observations over timescales (decades) and depths (sea surface to 40 m) relevant to understanding the oceanographic drivers of subsurface heating and their impacts on coral bleaching. Our analysis of long-term SST records indicates there have been 16 local-scale (0.1° × 0.1°) surface MHWs over the north shore of Moorea relative to a MMM-based bleaching threshold of 29.8 °C (Table S4), compared to 14 regional-scale (2° × 2°) events (Table S3). Localised heating over the north shore was often greater than regional SST would suggest, with the hottest event recorded in 2003 reaching 17.7 °C-days locally (Table S4), compared to 15.5 °C-days regionally (Table S3). Further details on historical events can be found in 'Surface MHW history around Moorea' in the Supplementary Information, which provides context for the six more recent surface MHWs that have occurred over the north shore since continuous in-situ, reef-level observations began at Moorea in 2005 (MHWs in 2007, 2012, 2015, 2016, 2017, and 2019; Table S4). Two recent, contrasting events

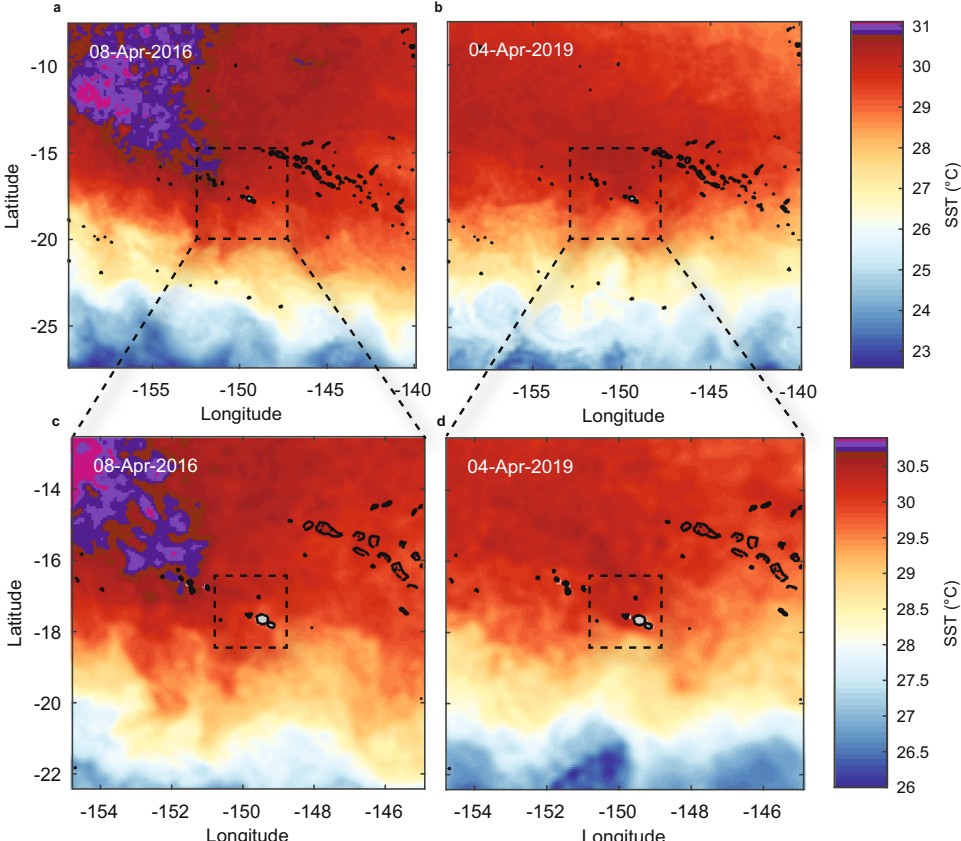

**Fig. 2 | Regional sea-surface temperature (SST) variability during the peak of the 2016 and 2019 surface marine heatwaves around Moorea suggested hotter conditions during 2016.** Panels show SST over **a, b** 20° × 20° and **c, d** 10° × 10° during 2016 and 2019, respectively, focusing on the date of peak SST observed over Moorea's north shore (8 April 2016 and 4 April 2019). Dashed squares in (**a, b**) show extent of (**c, d**) and those in (**c**, d) the extent of data shown in Fig. 1a, b (2° × 2°). Coastlines based on Wessel and Smith[77].

in 2016 and 2019 demonstrate the extent to which thermal environments at depth, and the associated severity of coral bleaching, can vary substantially from predictions based on sea-surface conditions.

MHW severity based only on SST can miss important information on the conditions experienced by organisms at depths greater than the surface skin layer quantified through remote sensing, which may only be few millimetres thick[34]. For example, although the localised peak in sea-surface temperatures were essentially identical between 2016 (30.1 °C; Fig. 1a) and 2019 (30.2 °C; Fig. 1b), and regional surface heating metrics and warnings were similar[33], markedly different heat accumulation occurred due to the different duration that temperatures remained above the putative coral bleaching threshold (MMM + 1 = 29.8 °C; up to 2 days in April 2016 compared to up to 11 days in April 2019; Table S1). Yet these results from local SST−of similar SST maximums in 2016 and 2019, but longer durations above the threshold in 2019−only capture some of the significant differences that led to constating MHW severity and ecological outcomes between years and across depths.

Daily average temperatures measured in situ at reef level in water depths of 10–40 m over ~15 years are well correlated with daily SST ($r^2$ = 0.94–0.78 at 10–40 m). However, the strength of the relationship between SSTs and in situ temperatures declines with increasing water depth, even when in situ temperatures are averaged to a daily resolution[31,35]. The potential for subsurface attenuation of heating over coral reefs has previously been demonstrated using high-resolution in-situ water temperature data in the context of both regional upwelling and local internal-wave climates[31,36,37], with observations of periodic transport of deeper, cooler water onto reef habitats at a large number of reefs globally[31,35,38]. The propagation of internal-wave energy is

associated with significant vertical displacements of density isopycnals and isotherms; e.g., ~60 m displacements along the Hawaiian Ridge[39]. Upon encountering a sloping bottom, internal-wave dynamics become complex, and, for habitats on the fore reef slope, typically result in rapid, periodic cooling (rather than oscillations around a mean temperature) as water masses associated with e.g., 24–27 °C isotherms are vertically advected onto the reef and recede again[31,40,41]. In deeper reef habitats there may also be periodic heating associated with exposure to warmer surface water masses when internal waves lead to downward displacement of isotherms[31], but the overall magnitude of any resulting net heating is small across the depth range considered here (i.e., no average heating at depths of 40 m and less; Fig. S4).

To separate the effects of low- and high-frequency processes driving heating across the reef slope, we used a filtering approach specifically designed and validated by Wyatt et al.[31]. to estimate coral reef thermal regimes without internal waves. In situ temperatures were filtered to isolate variability at frequencies higher and lower than the local inertial period (~40.0 h), effectively removing the effects of internal waves from lower frequency processes (i.e., multi-day weather patterns and seasonal effects; see Fig. S3). Contrasting the observed and filtered in-situ temperature variations (black and white lines, respectively, in Fig. 3) highlights differences in the processes driving the 2016 and 2019 subsurface MHWs. During the 2019 MHW around Moorea, the filtered, or 'non-internal wave' (NIW), temperatures closely resembled the observed temperatures (Fig. 3e–h, m–p) implying limited internal-wave cooling (IWC). Consistent warming across the water column was evident in 2019 and temperatures remained above the coral bleaching threshold for multiple days during early to late April (Table S1). By contrast, the 2016 MHW was characterized by temperatures remaining generally below the

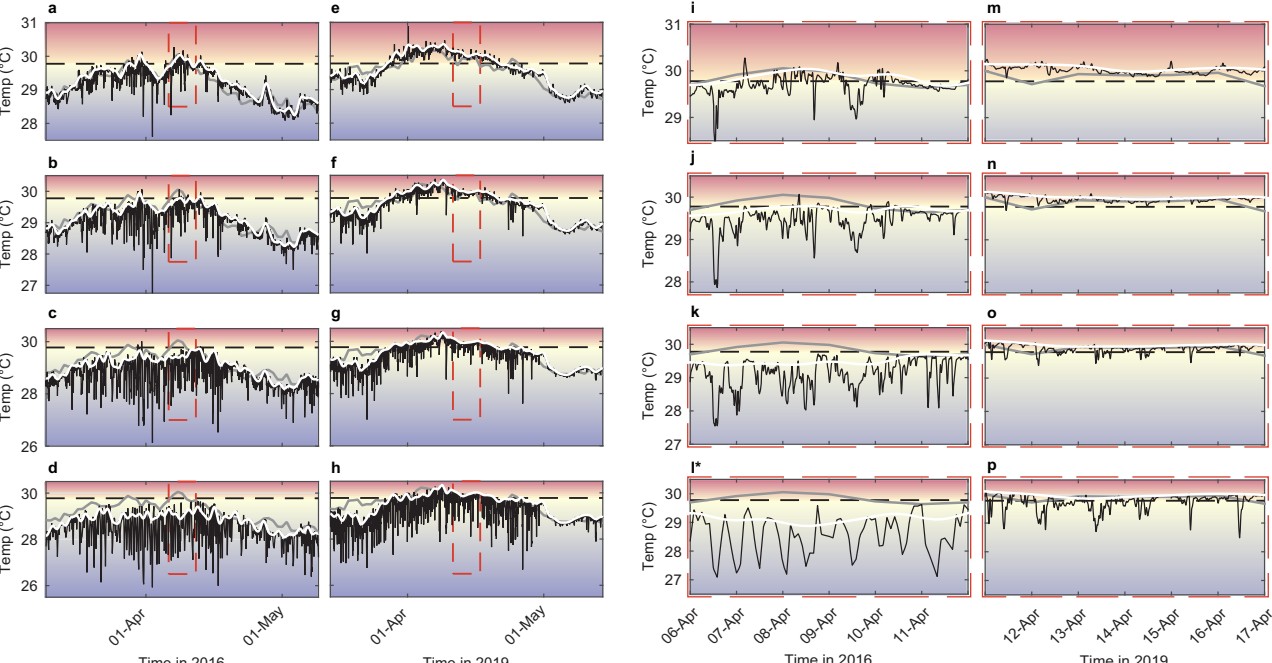

**Fig. 3 | Contrasting reef-level temperature variations across depths on Moorea's north shore during the 2016 and 2019 marine heatwaves.** Panels on the left show the observed high-frequency water temperature variations (black lines, measured at 2-min intervals) during the hottest months (Apr–May) in **a**–**d** 2016 and **e**–**h** 2019 at **a**, **e** 10, **b**, **f** 20, **c**, **g** 30 and **d**, **h** 40 m depths. Right panels focus on relative variation during the heatwave peaks across the same depths: **i**–**l**\* 06–12 Apr 2016 and **m**–**p** 11–17 Apr 2019. Non-internal-wave temperature variations are shown based on observed temperatures filtered to remove the high-frequency influence of internal waves (white lines). The satellite-derived sea-surface temperatures (SST; grey line) are shown for comparison to in situ temperatures. The horizontal dashed line shows the 'bleaching threshold' (maximum monthly mean + 1 °C) and the background shading provides a reference relative to temperatures above (red), equal (yellow) and below (blue) this threshold. The red dashed squares denote the axis limits in the right panels. \*Note: 40 m logger during 2016 incorrectly recorded at 2-h interval.

bleaching threshold and significant high-frequency variability indicative of IWC across depths, such that temperatures only exceed the predicted bleaching threshold for hours or less at a time (Fig. 3a–d, i–l; Table S1). The high-resolution temperature observations show that IWC was greatly reduced during 2019 (Fig. 3e–h). The power spectral density of observed temperatures, concentrated at semi-diurnal frequencies and consistent with internal-wave forcing at this location[41], was significantly higher in 2016 and lower in 2019 than the average across years at 10 m depth (Fig. 4a; see inset). In deeper water at 20–40 m depths, temperature variance within the semi-diurnal frequency band increased relative to shallow depths and became more similar between the two events, such that at 40 m the semi-diurnal variability was equivalent in 2016 and 2019 (Fig. 4b–d; see insets). However, this similarity in temperature variance does not indicate an equivalent magnitude of IWC, since variability during 2019 (Fig. 3p) was around a warmer background temperature closer to the coral bleaching threshold. Extending the comparison of 2016 and 2019 to other recent local MHWs demonstrates two distinct types of events: greater IWC across reef depths during 2012, 2015, and 2016 MHWs, versus reduced IWC during the 2007, 2017 and 2019 MHWs (Fig. 5).

The magnitude of temperature fluctuations produced by IWC, i.e., occurring at the semi-diurnal frequency, are of similar magnitude to long-term ocean warming and climate change threatening coral reefs globally. The average IWC ($\overline{IWC}$) during the high-IWC MHWs (2012, 2015 and 2016) was between 0.14 and 0.60 °C (Fig. 3a; Table S4) and comparable to the overall SST increase measured over tropical coral reefs during the last four decades (-0.65 °C[18]). As a result of the subsurface cooling caused by internal waves, the 2016 MHW, which was moderate at the surface (5.7 °C-days), was mild at 10 m (<3.2 °C-days) and negligible (<0.8 °C-days) at 20–40 m (Table S1). Peak IWC of between 0.3 and 3 °C regularly exceeds a level likely to be physiologically significant in

reducing stress responses such as coral bleaching during the hottest months (Table S5). IWC however is inherently temporally variable, with periods of both high and low IWC coinciding with MHW events (Fig. S4). Due to declining IWC during the development of the 2019 MHW, degree cooling days due to internal waves ($DCD_{IW}$) was between 36 and 60% less than during the 2016 event (Table S1), with $\overline{IWC}$ during the 2019 event averaging only 0.070 °C and 0.31 °C at 10 m and 40 m, respectively (Fig. 5b; Table S5).

The coincidence of reduced IWC with the peak of the 2019 MHW meant that, while maximum surface temperatures were similar to those in 2016, temperatures were elevated for a significantly longer period in 2019. This led to severe heat accumulation at the surface (15 °C-days) extending across depths (16.6–8.3 °C-days at 10–40 m) and persisting for days to weeks (Table S1). As a result, during the 2019 MHW, subsurface heat accumulation across depths was five- to 150-fold greater than in 2016 ($DHD_{max}$, Table S1).

Although IWC was significantly reduced in 2019 relative to 2016, temperature variability associated with internal waves was not entirely absent in 2019. For example, internal waves and associated IWC were evident in early 2019, especially at depths >20 m (Fig. 3g, h). The peak IWC at 40 m in 2019 was of similar magnitude to 2016 (3.09 °C and 2.93 °C, respectively; Table S5). Internal waves remained evident in deeper water during most of 2019 (Fig. 3g, h), with semi-diurnal variance at 40 m depth equivalent in magnitude between 2016 and 2019 (Fig. 4d). However, a key feature of the 2019 MHW, and likely those in 2007 and 2017, was the coincidence of limited internal waves, as evidenced by decreased $\overline{IWC}$ (Fig. 5b) and $DCD_{IW}$ (Fig. 5g, j), specifically during the periods of maximum SST (Fig. 1d). Unlike the 2016 MHW, when significant IWC coincided with the hottest SST, IWC was low throughout the peak of the surface heating events in March 2007, January 2017, and April 2019, especially in water deeper than 10 m (Fig. 5b). Deeper reef

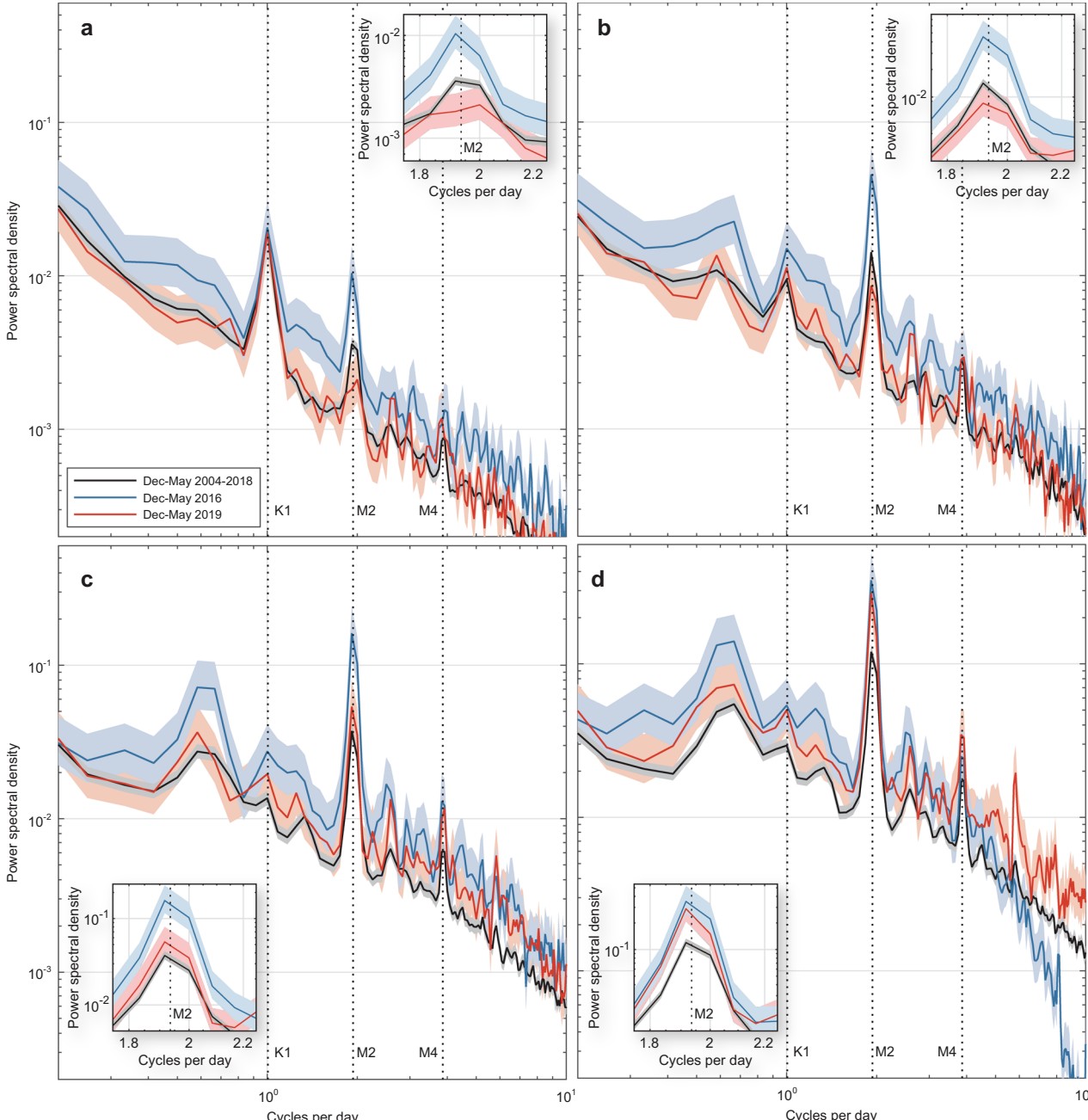

**Fig. 4 | Reduced semi-diurnal temperature variability during the summer of 2019 in shallower water on the north shore of Moorea.** Power spectral density (PSD) plots (logarithmic scale) were computed within a 12-day window at **a** 10 m, **b** 20 m, **c** 30 m, and **d** 40 m water depths during the summer months (Dec–May) in 2016 (blue), 2019 (red), and 2004–2018 (black; excluding 2016 and 2019). Shading shows the 95% confidence intervals for each PSD. The tidal constituents (dotted lines) show variance consistent with semi-diurnal (M2) forcing across depths, with diurnal (K1) forcing in 10 m of water along with some variability consistent with the shallow water lunar overtide (M4). Insets show details of semi-diurnal differences at each depth.

slope habitats generally exhibit more IWC than shallow habitats (Fig. S4) due to closer proximity to water column stratification (see below), and may therefore represent thermal refuges from MHWs[42]. However, the amount of cooling across depths, and thus refuge potential, appears to depend strongly on short-term processes influencing the vertical dynamics of water column stratification. Prolonged reductions in internal-wave arrivals could eliminate much of the refuge potential of deeper reef habitats as was evident during 2019 at Moorea.

## Oceanographic drivers of subsurface heatwaves
The majority of coral reef habitats globally are likely to be impacted by internal waves and resulting IWC, except for inshore reefs with limited exposure to the open ocean[31]. However, evidence of IWC may be lacking (i) due to analysis of in-situ temperature data at coarse temporal resolutions (e.g., daily averages), (ii) during seasons with limited water column stratification, or (iii) when mesoscale processes, including current meanders and mixing due to seasonal winds and large storms, lead to depression of the thermocline[31]. Such oceanographic processes seaward of reef slopes can strongly modify the mean depth of the thermocline and, thus, reef slope internal-wave climates. For example, along the ~180 km Florida Reef Tract, meanders and transient flow reversals of the offshore Florida Current strongly modify internal-wave activity detected on reef slopes[43,44]. Interactions of the Kuroshio Current with the Luzon Strait affect the westward

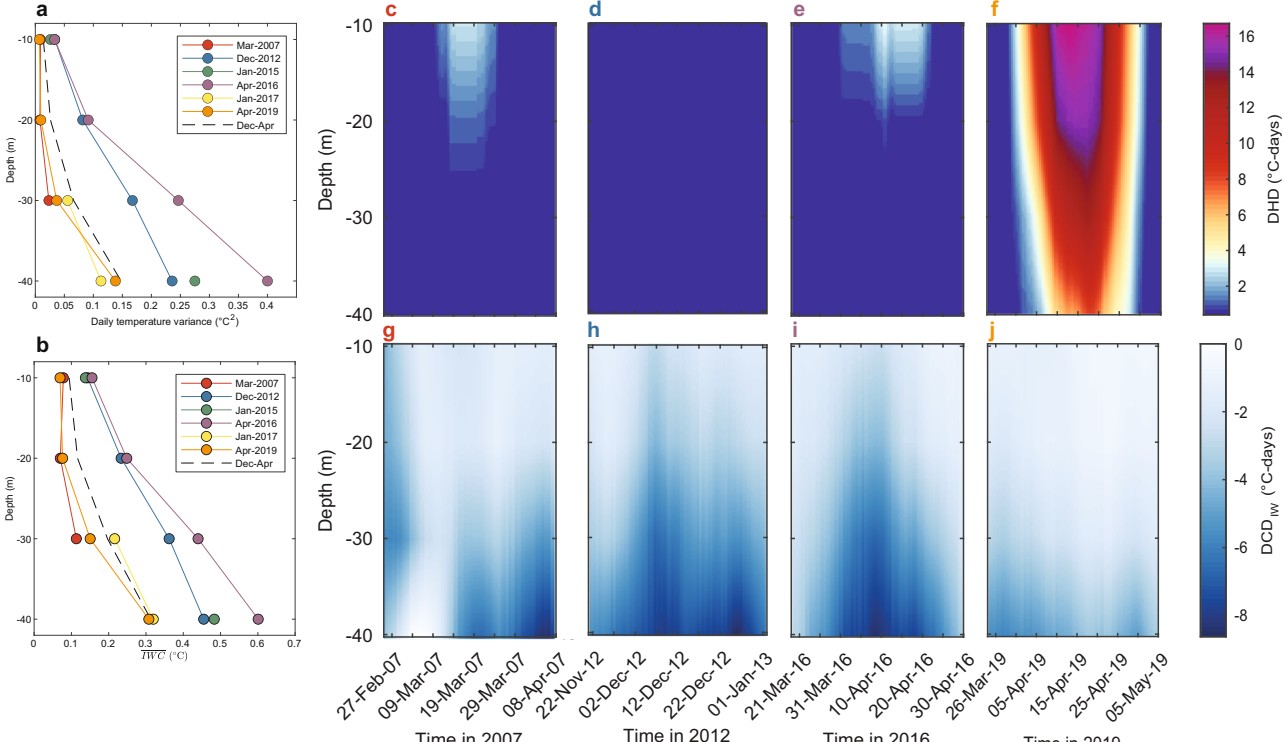

**Fig. 5 | Comparison of internal-wave cooling (IWC) across depths during recent surface marine heatwaves (MHWs) around Moorea.** Based on average **a** daily temperature variance (in °C²) and **b** IWC ($\overline{IWC}$ in °C) during the six recent local MHWs (see Table S4 for dates) that can be grouped into: (1) high IWC events during 2012 (blue), 2015 (green) and 2016 (purple); and, (2) low IWC events that coincided with bleaching events in 2007 (red) and 2019 (orange), along with early 2017

(yellow). The daily variance and $\overline{IWC}$ during Dec–Apr across all years (2005–2019) is shown for reference (black dashed lines). Contours of **c–f** heat accumulation as degree heating days (DHD in °C days) and **g–j** degree cooling days due to internal waves (DCD$_{IW}$ in °C days) across depths are shown for 2007, 2012, 2016 and 2019. Due to data gaps in the in situ records, contours are not shown for the 2015 or 2017 events.

propagation of internal waves into the northern South China Sea, which can alter the internal-wave climate over reef communities around Dongsha Atoll[45]. Moorea is also located within a dynamic oceanographic environment, characterised by the westward flow of the South Equatorial Current (SEC) and seasonal atmospheric migrations of the Intertropical Convergence Zone[46]. Despite the potential interactions, mesoscale oceanographic variability, and prevailing internal-wave climates, have not previously been considered in MHW or coral bleaching assessments.

Our analysis of remotely sensed dynamic sea level anomalies (SLAs) and SLA-derived surface currents near Moorea indicate contrasting patterns that are also consistent with the differences in heating observed across depths during the 2016 and 2019 MHWs (Fig. 6c, d). SLAs can be a good measure of fluctuations of the thermocline depth (taken as, e.g., the 20 °C isotherm depth) in the tropical Pacific Ocean, especially within ±15° of the equator where temperature stratification approximates a two-layer system and increased SLAs are linked to thermocline deepening[47]. For locations further from the equator including Moorea (~17.5 °S), stratification extends across ~200 m or more and is not well characterised as a two-layer system. In these locations the relationship between SLA and thermocline depth is weaker, although the change in thermocline depth for a given change in SLA may be greater. Rebert et al.[47]. suggested a 10 cm SLA increase corresponds to a ~38 m increase in the depth of the 20 °C isotherm based on sea level observations at Papeete and temperature profiles near Tahiti. These predictions are consistent with our observations of a depression of offshore isotherms and reduced IWC during the 2019 MHW (Fig. 7a; Fig. S6). The average depth of the 29.8 °C isotherm corresponding to the predicted bleaching threshold was 14.9 m (range 10.1 to 23.5 m) during

2016 (Fig. 1c), but was nearly double that depth during 2019, averaging 27.7 m (range 10.6 to at least 40 m; Fig. 1d). The depth of the 26 °C isotherm shows similar deepening in 2019, relative to both 2016 and to its long-term mean depth (Fig. 7a). During early 2019, the thermal structure offshore of Moorea in deep water (more than 1500 m) showed isotherm deepening in excess of that typical even during winter conditions (Fig. 7a)[31]. In early 2019 the 26 °C isotherm was at its deepest (max 139 m) observed since 1977 (data not shown), except for the winter of 2015 (max 140 m in late Aug 2015). It was on average 41 m deeper during April 2019 (130 ± 1.2 m), than in April 2016 (89.0 ± 1.2 m) (Fig. 7a; Table S3). Stratification was also weaker and deeper in Argo profiles north of Moorea during 2019 (Fig. 8a), with the depth of the maximum buoyancy frequency (N) on average 40 m deeper during 2019 (99.9 ± 51 m) than in 2016 (62.0 ± 9.9 m; Fig. 8b).

Corresponding with deeper stratification, SLAs measured in situ over the reef on Moorea's north shore were ~5 cm higher during Mar–Apr of 2019 (8.3 ± 0.2 cm) than 2016 (3.0 ± 0.3 cm; Fig. 7b) and SLA elevations offshore were associated with deeper maximum N in 2019 profiles (Fig. 8b). Reef-level SLAs during the 2019 MHW peaked at 11 cm and averaged 9.4 ± 0.1 cm (Fig. 7c; Table S4). By contrast, reef-level SLAs averaged 4.7 ± 0.3 cm during the 2016 MHW, 4.7 cm lower than during the 2019 event (Fig. 7c; Table S4), which is consistent with the isotherm depth predictions and observations discussed above. Depressed SLAs along with shallower maximum N (Fig. 7a; Fig. 8b) is consistent with the high levels of IWC measured on the reef slope during the 2016 MHW, whereas the elevated SLAs and deeper maximum N corresponded with limited IWC and prolonged heating at depth on the reef slope during the 2019 MHW.

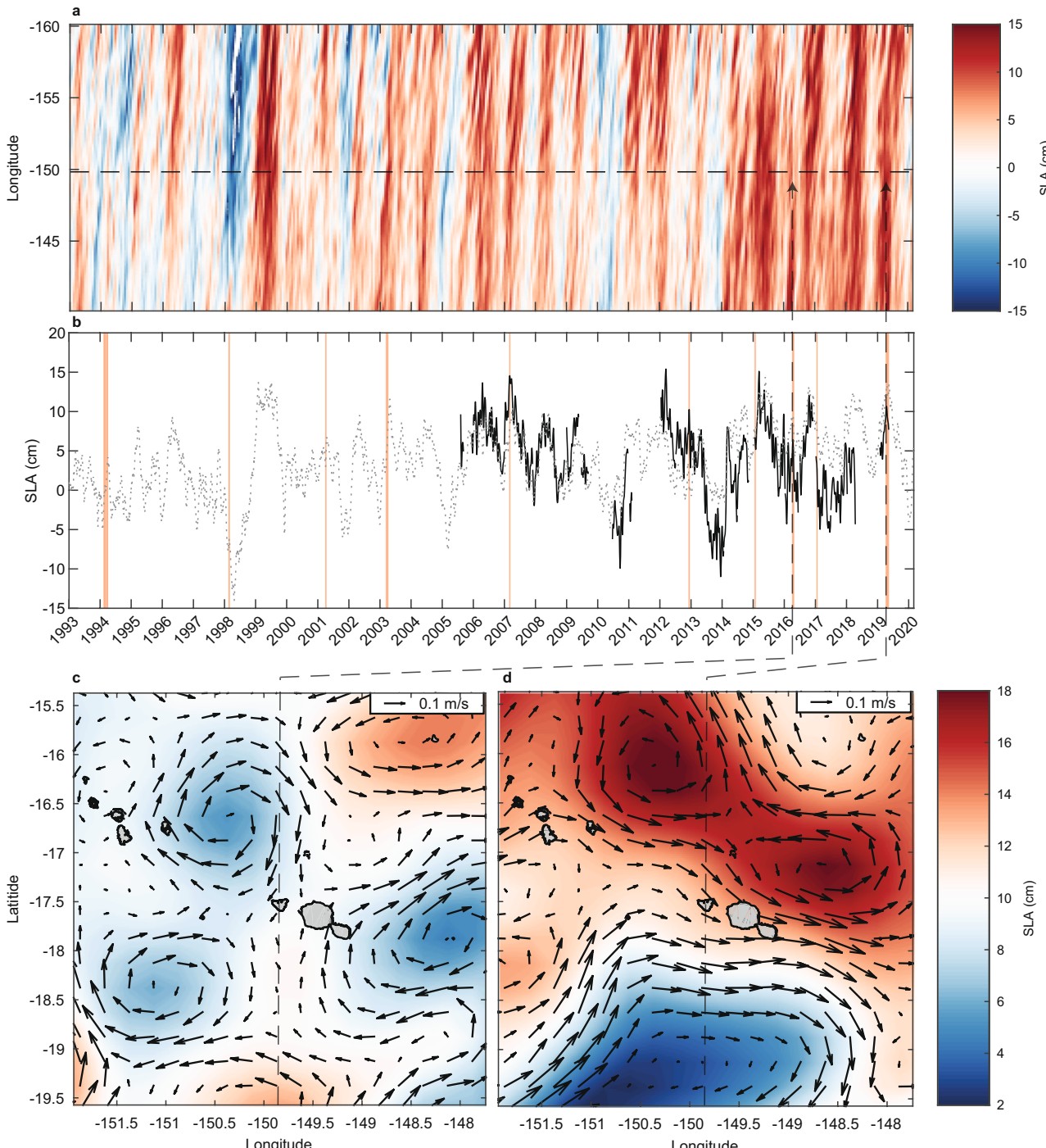

**Fig. 6 | The propagation of mesoscale sea level anomalies (SLAs) around Moorea highlighting contrasting eddy fields during the 2016 and 2019 marine heatwaves (MHWs).** Satellite altimetry shows **a** the long-term trends in SLAs along the latitude of Moorea (−17.5°; horizontal dashed line indicates longitude of −149.83°) and **b** changes in SLAs north of Moorea (dotted grey line) relative to in-situ SLAs measured at 10 m depth over the north shore fore reef (solid black line). **c** Lower SLA and a cyclonic eddy north of Moorea in 2016 contrasts with **d** higher SLAs and an anticyclonic eddy to the north of Moorea in 2019. Local surface MHW events (see Table S4) are shown in **b** (orange shading). Dates in **c**, **d** correspond to the peak SLA measured over the north shore reef slope during the two most recent MHWs (12 April 2016 and 16 April 2019, respectively; see Fig. 7c). Note there is a colour scale difference between **a** and **c**–**d** due to differences in spatial and temporal coverage. Coastlines in **c** and **d** based on Wessel and Smith[77].

With the benefit of longer-term data we can also explore in-situ links between SLA and IWC using records since 2005. The strongest IWC observed on the reef occurred in Dec 2013 (Fig. S4b) and coincided with the greatest depression in sea level observed since the 1998 El Niño (−11 cm; Fig. 6b). Conversely, some of the most elevated SLA observed over the reef occurred during the 2007 MHW (+14 cm; Fig. 6b and Table S3) when IWC was at similarly low levels to those observed during the 2019 MHW (Fig. 5b). During winter, the thermal signature of internal waves is minimal on the reef at depths shallower than 50 m because the mixed layer is deeper and temperature stratification associated with the thermocline occurs well below reef communities considered here[31,41]. Consequently, SLA depressions during winter do not appear to be associated with increased IWC (e.g., SLA of −9.9 cm and $\overline{IWC}$ of <0.075 °C at 10 m during Sep 2010; Fig. 6b and Fig.

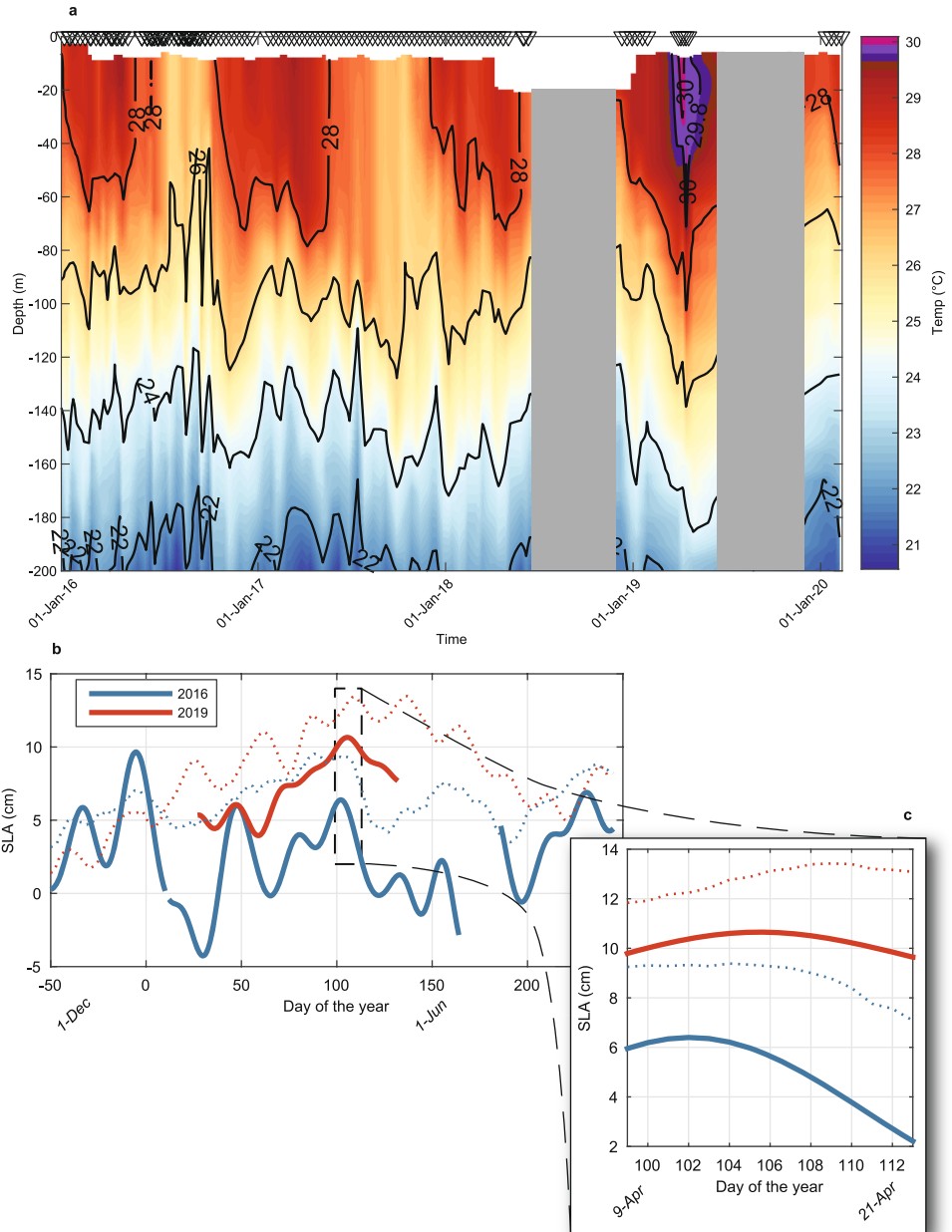

**Fig. 7 | Evidence of isotherm deepening and sea level elevation around Moorea during the 2019 subsurface marine heatwave. a** Broad-scale vertical thermal structure based on Argo and WOD casts around Moorea highlight isotherm deepening (black lines shown for every 2 °C) and surface layer heating during early 2019, with the coral bleaching threshold (29.8 °C) visible in the surface 50 m (periods without casts are greyed out). Comparison of **b** the sea level anomaly (SLA in cm) measured over the north shore reef slope at 10 m depth in 2016 (solid blue line) and 2019 (solid red line), **c** highlighting the relative elevation during April 2019. Data in **a** are smoothed by a weekly moving average to increase isotherm visibility. Dotted lines in **b**, **c** show average satellite SLAs north of Moorea (see Fig. 6) for comparison to the reef-level SLAs (solid lines).

S4a, respectively). By contrast, during the summer, the mixed layer is typically less than ~50 m deep[31], stratification within the thermocline is more strongly developed (Fig. 8a), and temporal changes in IWC appear to be closely linked to SLA (Fig. 8b).

Our analyses show that reef-level IWC, and thus temperature variance and overall heating of reef slopes, are dynamic among seasons and modulated within the summer season by mesoscale eddy dynamics embedded in the regional current system. Propagation of SEC eddies past Moorea are a potential source of SLAs, and corresponding variability in thermocline depths, at time scales of days to weeks. IWC would be expected to increase during the passage of cyclonic (clockwise in the southern hemisphere) eddies and depressed SLAs (Fig. 6c), and to be reduced under anticyclonic (counter-

clockwise) eddies and elevated SLAs (Fig. 6d). The rightward tilt in SLAs when plotted across longitude and time (Fig. 6a) suggests an overall westward propagation of SLAs across the central South Pacific and past Moorea, consistent with westward currents of the SEC within the northern limb of the South Pacific subtropical gyre (see SLA data animations in Supplementary Information).

Links between reef-level conditions and the dynamics of mesoscale currents and eddy fields further demonstrates that any potential refuge from heating associated with deeper reef habitats[6,31,48] could be altered, and potentially eliminated, by mesoscale changes such as thermocline depressions, as well as with increases in thermocline depths expected with longer term ocean warming and increasing stratification[49]. A general increase in SLAs near Moorea beginning in

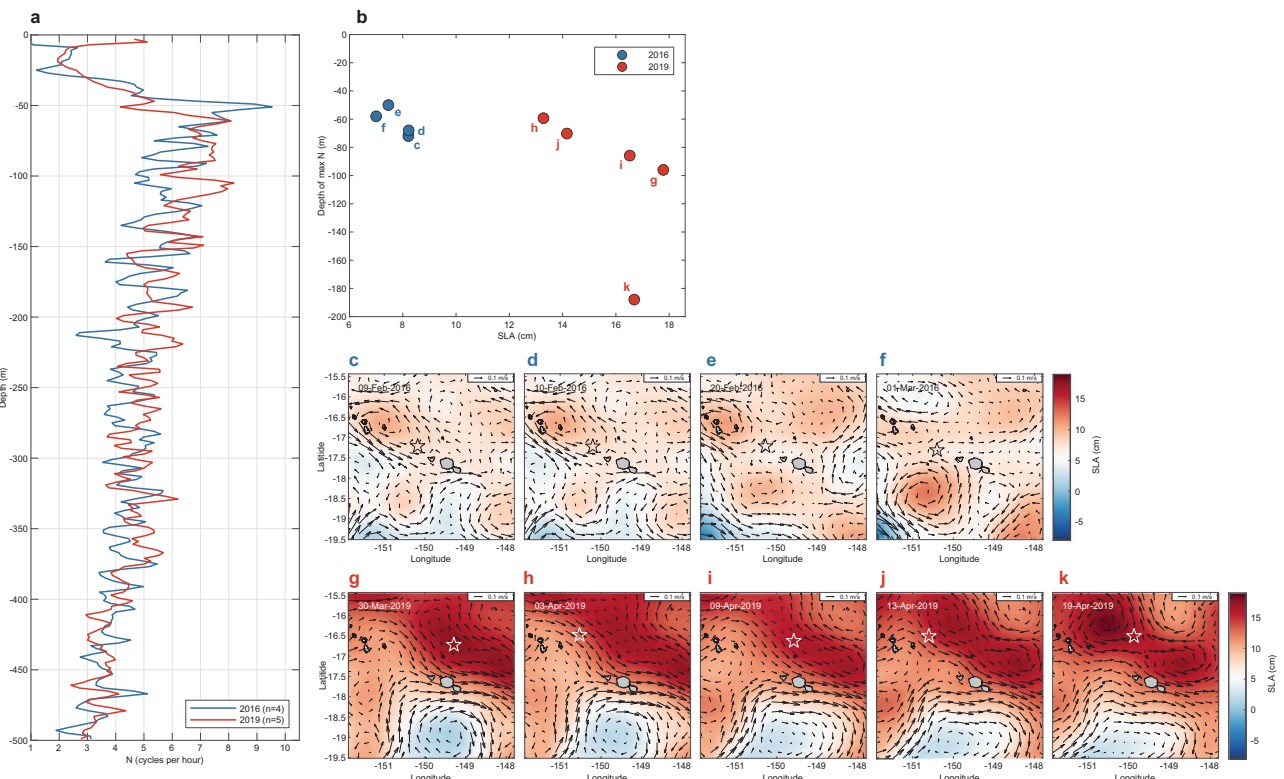

**Fig. 8 | Propagation of warm-core eddies north of Moorea reduced stratification and increased mixed-layer depths during the 2019 marine heatwave (MHW).** Vertical profiles of the buoyancy frequency (N, cycles per hour) are shown **a** averaged over four and five Argo profiles (2 m depth bins) for early 2016 (blue line) and 2019 (red line), respectively, suggesting a shallower and more pronounced stratification (N peak) during 2016. **b** Elevated sea level anomalies (SLA, in cm) over Argo profiles during early 2019 (red circles) were associated with deeper mixed-layer depths (depth of max N, in m; 99.9 ± 51 m) than in 2016 (62.0 ± 9.9 m). Opportunistic Argo profiles around Moorea only occurred during (**c**–**f**) the pre-MHW, moderate SLA depression period prior to the 2016 MHW (black stars) but captured (**g**–**k**) the peak of the 2019 MHW and SLA elevation due to warm-core eddies (white stars).

2014 (Fig. 6a) may be evidence of increasing thermocline depths, with consequently decreased IWC and more frequent coral bleaching in the future. However, during the 2015–2016 El Niño, this increase was interrupted by lower SLAs passing Moorea, as indicated by the white banding in Fig. 6a during 2016. An ongoing trend of increasing mesoscale variability due to eddies[50] could be expected to increase variability in internal-wave climates and the severity of subsurface MHWs over coral reefs in many regions.

## Ecological implications of climate change and thermocline dynamics

Subsurface alteration of the intensity and scale of MHWs by mesoscale processes can have important ecological impacts on marine ecosystems including coral reefs. We highlight that reliance on SST data over coarse spatial (10 s km² or more) and temporal (weeks) scales in order to correlate MHWs with ecological effects through inferred, or occasionally empirical, effects on physiological stress can generate misleading conclusions. In the case of Moorea during 2016 and 2019, regional stress estimates based on daily SST data would support predictions of moderate bleaching in both years, at least when gauged against the SST-derived bleaching threshold for this region (MMM + 1 = 29.8 °C). However, directly observed coral bleaching on Moorea's north shore fore reef differed markedly from this prediction.

In contrast to predictions based on SST, there was only minor bleaching in 2016 that was short in duration and restricted to shallow depths, whereas there was a severe and prolonged bleaching event from April to May 2019 that was anecdotally reported as occurring to 100 m depth[51]. The five-fold increase in the peak heating recorded at 10 m water depth in 2019 relative to 2016 (16.6 and 3.22 °C-days,

respectively; Table S1) coincided with qualitative measures of coral bleaching (coral colour in 36 to 40 photo quadrats per site and depth) that affected only 11.5 ± 1.9 to 17.4 ± 2.9 % (mean ± S.E.) of live coral cover at 10 m depth in early 2016 (10 May), but 54 ± 8.6 % in early 2019 (1 May) (Fig. 1g, h; see Supplementary Information for details). Bleaching was even more severe in 2019 when assessed as the proportion of colonies within the dominant *Pocillopora* genus exhibiting bleaching (71–72%), with most colonies (55–65%) severely bleached[27]. Bleaching occurrence however provides little information on the ultimate ecological impact of a MHW.

Coral mortality following bleaching is an important aspect of the ecological impact of extreme ocean temperatures, with mortality expected to be more widespread as thermal stress events caused by MHWs increase in frequency and spatial extent[19]. The 2016 MHW was not a source of ecologically meaningful coral mortality[52], with no detectable change in coral cover at either 10 m or 17 m depth on the fore reef by August 2016 (Fig. 9e). As a result, the event did not attenuate the upward trajectory in coral cover evident on the fore reef since 2012[53]. By contrast, consistent with our observations of a prolonged subsurface MHW in 2019 (Table S1), bleaching led to significant coral mortality, estimated to be as high as 42% for the ecologically dominant coral *Pocillopora* spp. by August 2019[27]. While coral abundance at Moorea is highly dynamic at multi-year scales, mortality during the MHWs recorded prior to 2019 seems to have been minimal[54] (see 'Historical bleaching at Moorea' in Supplementary Information). Between 2011 and 2019, coral communities around Moorea had undergone nearly a decade of recruitment and rapid growth (e.g., live cover reaching 67 ± 1.7 to 76 ± 1.8% at 10 m depth on the north shore by April 2019; Fig. 1g, h; Fig. 9). This increase in coral cover followed a severe outbreak of *Acanthaster planci* (crown-of-thorns

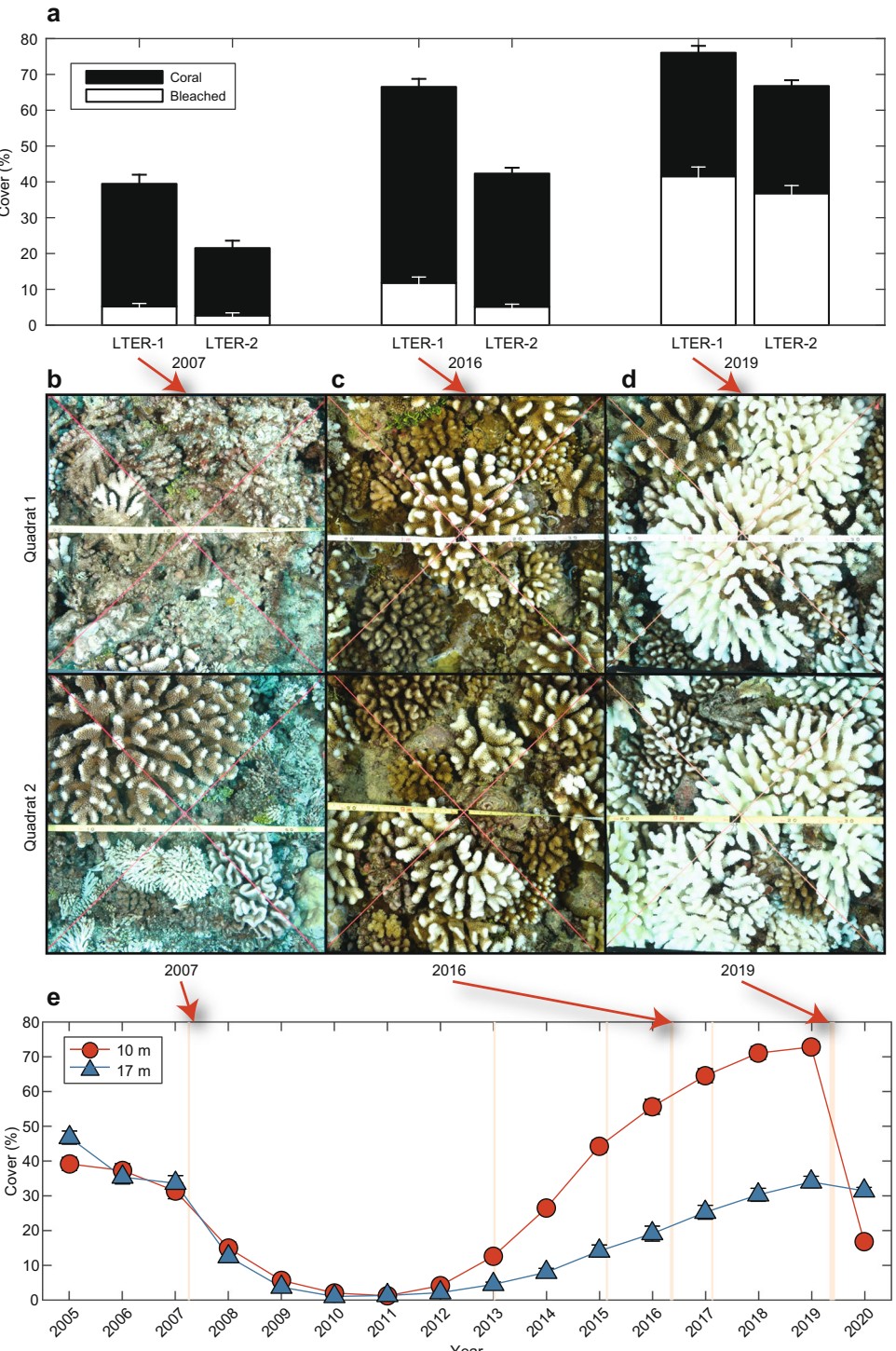

**Fig. 9 | Increased prevalence of coral bleaching and subsequent mortality associated with the hidden 2019 subsurface marine heatwave (MHW) following almost a decade of recovery in live coral cover on Moorea's north shore fore reef.** Bars in **a** show the cover (in % area) of coral (black) and bleached coral (white) observed at 10 m depth at two Long-Term Ecological Research (LTER) sites (LTER-1 and LTER-2) during the 2007 (20 Apr), 2016 (10 May) and 2019 (1 May) MHWs. Values are mean ± SE ($n$ = 36–40 quadrats) for all corals (pooled among taxa). Images (**b**–**d**) show examples of the coral community at each of these time points, respectively, at 10 m depth at LTER-1. **e** Live coral cover, averaged across LTER-1 and LTER-2, declined by -56% at 10 m water depth (red) compared to -10% at 17 m (blue) after the 2019 MHW. Local surface MHW events (see Table S4) around Moorea are shown in (**e**) by vertical orange shading.

starfish) first observed around 2004[55] and a major cyclone in 2010, which reduced live coral cover to close to zero at many sites[9] (Fig. 9e). High rates of coral recruitment are an important aspect of the resilience of Moorea's coral communities on the fore reef[9], along with the co-occurrence of cryptic species within the dominant coral genus *Pocillopora*[27,28]. Our observations of the subsurface severity of the 2019

MHW, with widespread coral bleaching and subsequent high coral mortality, highlight the potential significance of mesoscale eddies and SLAs in modulating thermocline dynamics and altering coral community mortality and resilience.

A similar interaction between mesoscale eddies and sea level dynamics, IWC and bleaching in Moorea may have occurred during the

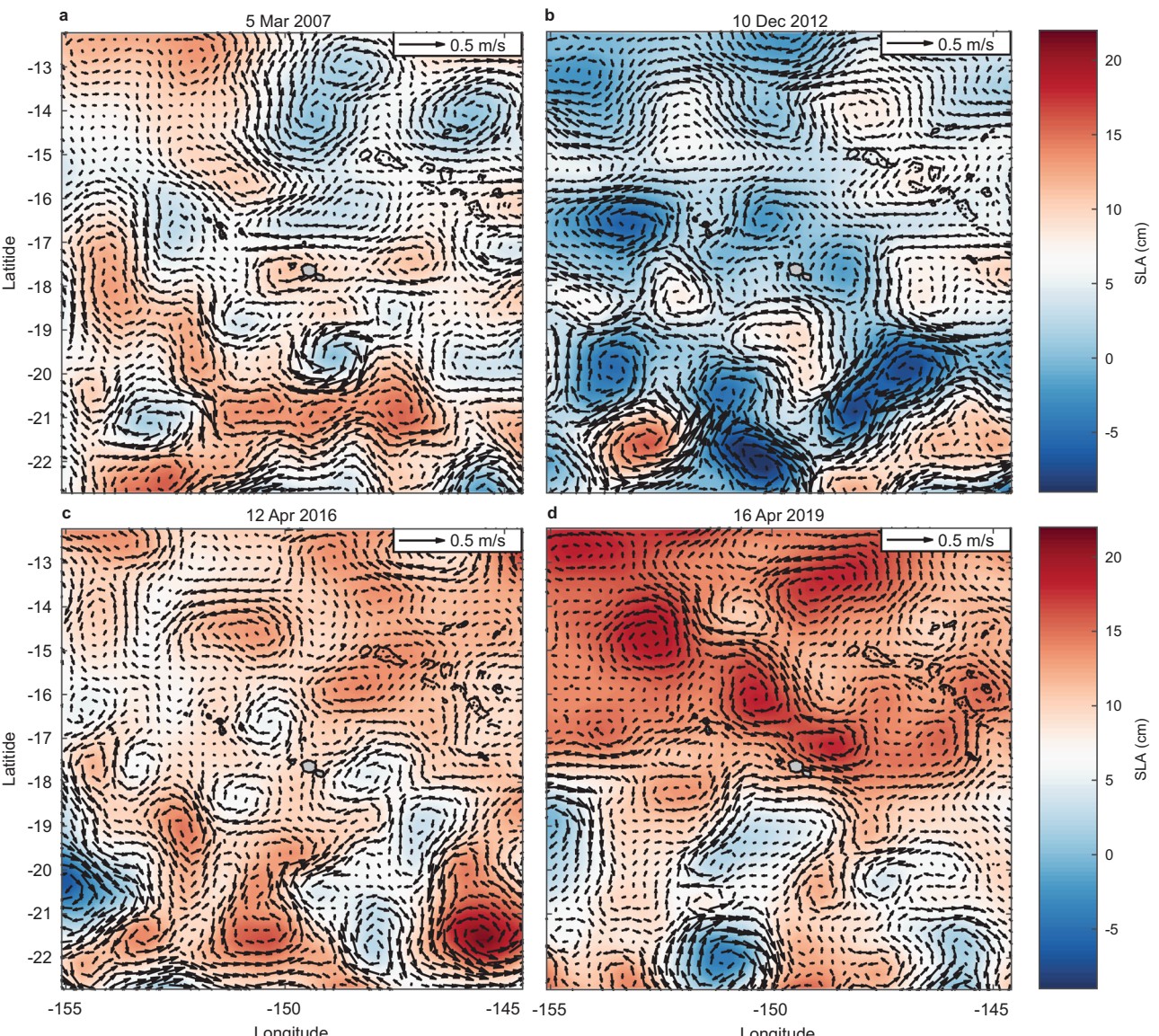

**Fig. 10 | Comparison of regional mesoscale sea level anomalies (SLAs) and eddy fields around Moorea during the surface marine heatwaves (MHWs) of 2007, 2012, 2016 and 2019.** Plots show SLAs from satellite altimetry on **a** 5 Mar 2007, **b** 10 Dec 2012, **c** 12 April 2016, and **d** 16 April 2019, highlighting the **a** elevated SLA around Moorea during the minor subsurface MHW in 2007, **b** regional SLA depression during the surface-restricted MHW in 2012, **c** lower regional SLA and a cyclonic eddy north of Moorea during the moderate subsurface MHW in 2016, and **d** extensive region of higher SLA and anticyclonic eddies to the north of Moorea during the severe subsurface MHW in 2019. Dates correspond to the peak SLAs measured over the reef during each MHW (e.g., see Fig. 7c for 2016 and 2019). Coastlines based on Wessel and Smith[77].

2007 MHW. Both surface and subsurface heating were relatively mild during this event (Fig. S7c), but bleaching across the reef was suspected to "have been exacerbated by calm sea conditions and associated increases in solar exposure, especially in deeper habitats"[54]. Because this event was coincident with a major population outbreak of *A. planci*[55], it is difficult to determine how much of the precipitous decline in coral cover after 2007 was caused solely by corallivory and how much may have been accentuated by bleaching mortality. During early 2007 there was evidence of elevated sea level (Fig. 10a; Table S3) associated with an anticyclonic eddy centred near Moorea (Fig. S7e), and low IWC comparable to that recorded in April 2019 (Fig. 5b), leading to subsurface heating consistent with that recorded by satellite over the north shore (2.2 °C-days; Fig. S1). In-situ heating (i.e., recorded with loggers) in 2007 reached 2.72 °C-days and 1.1 °C-days at 10 m and 20 m depth, respectively (Fig. S7). As in 2019, reduced IWC associated with elevated sea levels suggests the possibility that a subsurface MHW

also contributed to higher-than-expected bleaching severity during the 2007 event and subsequent loss of coral cover through bleaching-induced mortality.

There is little evidence that cold spells associated with increased IWC, and potentially nutrient upwelling, induced coral bleaching on the fore reefs of Moorea. There was no bleaching recorded during the strong IWC event of 2013–2014 to suggest a cold water or nutrient effect (Fig. S4). A lack of bleaching in our transects does not conclusively indicate the absence of bleaching across the reef, and we do expect biogeochemical effects in the water column from propagating eddies[56,57]. However, based on the long-term trajectories in coral recovery during this period[9] (Fig. 9e), coupled with our observations of less bleaching than expected in 2016 despite increased internal-wave upwelling, it seems clear that there was little or no ecological relevant bleaching mortality on the fore reef of Moorea related to nutrients upwelled during these events (e.g., <1% mortality[52]). Further, given

clear difficulty in predicting subsurface heating with remote sensing, it is not surprising that attempts to explain bleaching events using SST data alone[58], especially those that focus on coarse-resolution weekly DHW (Fig. S5), may not find strong evidence of the effects of elevated temperature on rates of bleaching. Quantitative assessments of MHW impacts require high-resolution, in-situ temperature observations that incorporate the influence of mesoscale eddies, changing thermocline dynamics, and internal waves. In addition to cooling effects, future research could examine the potential role of internal waves in bleaching mitigation through upwelling of organic particles associated with the deep chlorophyll maximum[31,40,59,60], which could enhance coral feeding and also improve outcomes for thermally stressed corals[61–63].

### Internal-waves as a component of changing ocean weather

Most ocean-exposed coral reef slopes are exposed to shallow summer thermoclines that intersect the coast within the depth range of reef communities, leading to at least some, and potentially extensive, internal-wave cooling across depths during the hottest months of the year when water column temperature and density stratification are maximal[31]. Here we have demonstrated that there can be mechanistic links between mesoscale eddies and thermocline dynamics that lead to increased water column heating and decreased internal-wave cooling. Subsurface MHWs, as observed in 2019, can lead to rapid coral bleaching and subsequent mortality across broad depth ranges. Moorea's internal wave climate is, however, not remarkably energetic on a global scale, with many reef locations likely experiencing greater internal wave exposure due to a variety of factors, including stronger or shallower stratification and closer proximity to internal wave generation sites[31]. Oceanic eddies associated with regional currents such as the SEC may be a dominant feature of surface conditions influencing coastal ecosystems[50]. Our observations at Moorea may thus represent the lower end of the impacts of mesoscale eddies on thermocline dynamics and internal-wave cooling of reef communities globally. Predicting the future impact of MHWs on subsurface coastal ecosystems depends on understanding mesoscale drivers of thermocline dynamics in the context of long-term climatic forcing. Increasing stratification and deepening thermoclines under climate change[49,64] are likely to reduce the potential for internal-wave refuges to protect coral reef communities from continued surface warming. We reiterate calls to consider the nature and extent of subsurface environmental fluctuations[35], broadly termed 'ocean weather'[65], as key factors in the response of coral reefs and other coastal ecosystems to ocean heating.

## Methods

### Sea-surface temperatures (SSTs)

Daily SST values were derived from satellite observations as described for regional SST in Wyatt et al.[31], but here focused on SST averaged over a smaller 0.1° × 0.1° square (four pixels, corresponding to a ~10 km × 10 km) centred on the north shore of Moorea in the Society Islands, French Polynesia (see Fig. 1a, b). Briefly, SST data were obtained for the period 1985–2019 using the 'CoralTemp' product produced by the National Oceanic and Atmospheric Administration (NOAA) Coral Reef Watch[29] based on Maturi et al.[66]. and Robert-Jones et al.[67]. The long-term climatological maximum monthly mean (MMM) SST value calculated for the Society Islands (2° × 2° around Moorea) was 28.8 °C[31]. Broader regional patterns in SST are provided in Fig. 2 and temporal evolution of the regional MHWs in 2016 and 2019 can be visualised in the data animations provided in the Supplementary Information. An analysis of SST across different spatial scales around Moorea is described in the Supplementary Information (Fig. S1).

### In situ temperatures

Long-term temperature records were collected at 2-min sampling intervals on the 10, 20, 30 and 40 m depth isobaths at Moorea between

Dec 2004 and Aug 2019 (Table S2) using Seabird Electronics SBE39 and SBE56 temperature recorders (0.002 °C accuracy, 0.0001 °C resolution, <10 s response time) bottom-mounted directly onto the reef surface, i.e., they measured water temperatures immediately overlying the coral communities of interest. Due to an instrument setup error, the 40 m logger incorrectly sampled at 2-hr intervals between 21 Aug 2015 and 25 Aug 2016. The horizontal distance between the loggers was about 40–100 m based on the reef slope at Moorea varying between 10 to 25%[41] (see Fig. S1e). Power spectra for these in-situ temperature records (Fig. 4) were calculated using the Welch average periodogram method following Emery and Thomson[68] and Trauth[69], as described in Leichter et al.[41]. Window length was set to 12 days (i.e., 8640 points of 2-min interval data) with application of a Hamming window and 50% overlap and calculation of 95% confidence intervals around the power spectral values.

### MHW quantification

Heat accumulation was quantified according to the convention for coral bleaching events[15], which is based on a degree heating concept. Most coral bleaching studies focus on Degree Heating Weeks (DHW, in °C weeks), which provides a metric of heat accumulation over the preceding 12 weeks. Here, to obtain heat accumulation at higher resolution and that was more temporally synchronous with the environmental conditions, we calculated Degree Heating Days (DHD, in °C days) as described in Wyatt et al.[31]. Briefly, DHD were calculated as the sum of temperatures (T, in °C at 2-min resolution) 1 °C or more above the MMM SST ($\hat{T} = T - 28.8$, in °C) within a 12-day moving window divided by the number of samples in each day (n = 720 for 2 min sampling interval data):

$$\text{DHD} = \frac{\sum_{t-12n}^{t} \begin{cases} \hat{T}(t) \; if \; \hat{T}(t) \geq 1 \\ 0 \; if \; \hat{T}(t) < 1 \end{cases}}{n} \quad (1)$$

The DHD estimates using a consistent SST-derived MMM for all depths are conservative since the SST-derived MMM is higher than in situ MMMs, e.g., by up to 2.7 °C relative to in-situ temperatures at 40 m at Moorea[31]. The extent to which coral bleaching thresholds are influenced by local variations in thermal environments (i.e., across depths) is unclear. There is evidence that bleaching responses could be relatively intrinsic[70] and thus may not change significantly with depth relative to a fixed threshold. Comparable estimates of DHW can be calculated in the same way using Eq. (1) but over a moving window of 12 weeks of weekly data[15]. The use of a 12-week window for DHW (and our analogous 12-day window for DHD) is somewhat arbitrary and was originally designed to be consistent with estimates of heating made in earlier studies that used monthly SST data (e.g., Degree Heating Months calculated over 3 months)[15]. DHW values above 4 °C-weeks have been considered likely to lead to significant bleaching and values above 8 °C-weeks are likely to lead to severe bleaching and widespread mortality[71]. To avoid complications associated with variations in the location-, habitat-, and species-specific relevance of fixed warning levels, we identified MHWs quantitatively based on surface heat accumulation (DHD above 1 °C-days) but events were compared without quantitative reliance on any specific bleaching warning levels.

### Internal-wave cooling (IWC)

As a means of comparing internal-wave activity between MHWs, we estimated the cooling caused by internal waves using a previously developed metric based on differences between an estimated non-internal wave (NIW) signal and observed (Obs.) temperature time series[31]. The goal here was to estimate what the temperatures on the reef slope would have been in the absence of internal waves observed over the slope and use those estimates to estimate the amount of internal-wave cooling (IWC). The NIW signal was derived according to

Wyatt et al.[31]. by first filtering the Obs. time series to separate the low- and high-pass signals (LP and HP, respectively) corresponding to frequencies lower and higher than the local inertial period (40.0 h). Because the reef-level temperatures were measured at depths shallower (10–40 m) than the depth of the offshore water column thermocline near Moorea (~50–200 m), the effect of internal waves is predominantly to cause rapid cooling. This is evidenced by downward deviations in temperature as isotherms are displaced upwards and onto the reef during the run-up of non-linear and broken internal waves on the reef slope, followed by a return to warmer background conditions as cooler water recedes off the reef[41]. We confirmed that internal waves led to cooling followed by warming back up to close to pre-event conditions by identifying the timing of major events in the time-series (e.g., Fig. S2 for 20 m), with cooling occurring on average (±S.E.) 2.2 (±0.2), 2.6 (±0.2), 2.2 (±0.1), and 2.0 (±0.1) hours, respectively, before re-warming at 10, 20, 30 and 40 m ($n = 8$, 32, 148, and 319 events, respectively, since 2016). This cooling and re-warming is different from symmetrical oscillations around a mean temperature likely to be observed in cases of linear interval waves causing upward and downward displacements of isotherms offshore, which might be observed far from the seafloor or reef bathymetry. In such open ocean settings in deeper water, linear internal waves may tend to cause oscillations of isotherms around a mean state, although non-linear dynamics and soliton waves of depression are often also observed. Unlike the case of symmetrical oscillations around a mean temperature, where the high-frequency deviations might average to approximately zero, in the case of reef-level temperatures the low-pass signal contains the effects of net cooling from internal waves interacting with the reef slope, which would tend to increase apparent internal-wave differences between periods with and without pronounced internal-wave activity. To remove this net cooling effect from the low-pass signal, we estimated the root-mean-squared of the high-pass signal, rms(HP), within a moving window the size of the local inertial period (40.0 hrs), and added this quantity to the low-pass signal to estimate the NIW signal; i.e., NIW = LP + rms(HP). The NIW signal is thus an estimate of temperature conditions free from the non-linear influence of internal waves on reef-level temperatures. See Fig. S3 for a visual demonstration of the non-linear, downwards influence of internal waves on reef-level temperatures and the NIW time series derivation described and validated in Wyatt et al.[31].

IWC, in °C, was then calculated at the same resolution as the observed data (here 2-min) by subtracting the observed temperatures from the NIW:

$$IWC = NIW - Obs. \qquad (2)$$

Since IWC is computed at the same high temporal resolution as the Obs. data, our temporal comparisons of IWC focus on the average IWC ($\overline{IWC}$) over time periods corresponding to specific MHWs (Fig. 5b and Table S5) or over a 12-day window for longer-term comparisons (Fig. S4). A cumulative measure of the cooling effect of internal waves was also calculated in an analogous way to the degree heating quantification above, by calculating Degree Cooling Days due to internal waves ($DCD_{IW}$) using Eq. (1) and substituting IWC for $\bar{T}$ and summing all values (including those <1 °C; Fig. 5g–j, Table S1).

Our method for calculating IWC represents an estimate of the effect of internal waves on the temperature time series and is especially useful for providing a consistent means of comparing thermal variability among events, such as the MHWs in different years. A caveat of the approach is that the addition of the rms(HP) would not be appropriate for estimating background, or NIW, temperatures in the case of symmetrical oscillations around a mean temperature, such as might be observed for linear internal waves near the depth of the pycnocline in deep water, e.g., from open ocean mooring data. For non-linear waves interacting with a reef slope, the rms(HP) addition

removes the cold bias in the LP signal, proportional to the variability at frequencies faster than the local inertial frequency, and results in a NIW signal that closely matches the higher temperatures observed between the non-linear IWC events.

## Coral bleaching prevalence

The extent of bleaching of the coral community during 2016 and 2019 on the north shore of Moorea was assessed along transects at two permanent sites that are part of the Moorea Coral Reef Long Term Ecological Research (LTER) program[9] as described in Edmunds (2018)[53]. Briefly, photoquadrats (0.5 m × 0.5 m) were recorded at each site (36–40 quadrats at each of 10 m and 17 m depth at LTER 1 and LTER 2 from 2005 to 2020) during April or May. Quadrats were analysed using CoralNet software[72] with full manual annotation with 200 randomly located dots on each colour image. The full analytical resolution separates corals (scleractinians and *Millepora*) by genus, but here the data are presented as overall coral cover (pooled among taxa), and the proportion of the live coral cover that was bleaching. Bleaching was recognized by complete white coloration beneath each dot. While we cannot be certain that some of the white areas were not caused by factors other than bleaching (e.g., *Acanthaster planci* predation or disease), these possibilities are considered rare since there was no active *A. planci* outbreak in 2016 or 2019 and areas of dead coral do not remain white for long as they are quickly colonized by algae.

## Regional hydrography

Regional hydrographic information in a 2° × 2° box around Moorea (Fig. 1a, b) was obtained from the NOAA World Ocean Database (WOD) and Argo. A total of 79 WOD CTD and 443 Argo casts were identified between Nov 1977 and Feb 2020, of which 350 (61%) were selected for further analysis based on having been deeper than 200 m. A total of 302 casts occurred since the in-situ observations began on the reef (Dec 2004).

## Sea level anomalies

Sea level anomalies (SLAs) were assessed based on both reef-level measurements and satellite altimetry data. Reef-level SLAs were computed from bottom pressure data recorded using SeaBird Electronics SBE26plus Seagauge Wave & Tide Recorders. Recorders were mounted on plates firmly attached to the reef. These recorded semi-continuously at ~10 m depth on the fore reef on Moorea's north shore (Fig. 6b). Atmospheric pressure changes were removed from bottom pressure records using barometric measurements from the Gump Field Station and the Moorea Temae Airport. Bottom pressure data were low-pass filtered with a cut-off frequency of 1/20 day$^{-1}$ and converted to water depth using water column measurements of seawater density. Satellite estimates of SLA were calculated from altimetry data available from 1993 to March 2020 from the Copernicus Marine Service[73]. Satellite SLAs north of Moorea were averaged over two grid points at the northeast and northwest corners of a rectangle (0.5° latitude × 1.5° longitude) centred on the island for comparison to reef-level SLA.

## Data availability

Satellite SST observations can be accessed at https://coralreefwatch.noaa.gov/satellite/index.php. In-situ data for the Moorea LTER site can be accessed at https://mcr.lternet.edu/data (seawater temperatures: knb-lter-mcr.1035.12[74]; sea level anomalies: knb-lter-mcr.30.35[75]; coral cover: knb-lter-mcr.4.38[76]). Satellite altimetry from the E.U. Copernicus Marine Service can be accessed at https://marine.copernicus.eu[73]. The NOAA World Ocean Database (WOD) can be accessed at https://ncei.noaa.gov/products/world-ocean-database and Argo data from https://ncei.noaa.gov/products/global-argo-data-repository.

## Code availability

The Matlab® script used to perform the internal-wave filtering described in this paper is available at https://github.com/OceanEcol/non-internal-wave (with identifier: https://doi.org/10.5281/zenodo.7321910) or from the corresponding author upon request.

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

## Acknowledgements

K. Sydel and C. Gotschalk assisted with access and initial processing of the core time-series data from the Moorea Coral Reef (MCR) LTER Site, which is funded by the US National Science Foundation (NSF) under Grant No. OCE-1637396 (and earlier awards) as well as a gift from the Gordon and Betty Moore Foundation. Research in Moorea was completed under permits issued by the French Polynesian Government (Délégation à la Recherche) and the Haut-commissariat de la République en Polynésie Francaise (DTRT) (Protocole d'Accueil 2005–2020). This work represents a contribution of the MCR LTER Site. A.S.J.W. was supported by funding from the Research Grants Council (RGC) of Hong Kong (RGC Project No. 26100120). J.J.L. was partially supported during the analysis and manuscript preparation by funding from NSF Grants No. OCE-1535203 and OCE-2022959.

## Author contributions

The study was conceived, and analyses carried out, by A.S.J.W. A.S.J.W. and J.J.L. wrote the initial drafts of the paper. L.W. and L.K. provided processed sea-level data, and P.J.E. and S.C.B. provided processed coral bleaching data. All authors contributed to subsequent analysis, writing and editing.

## Competing interests

The authors declare no competing interests.
