## [Peer Review File · Nature Communications]

Hidden heatwaves and severe coral bleaching linked to mesoscale eddies and thermocline dynamicsReviewer #1 (Remarks to the Author):

see attached document.

Reviewer #1 Attachment on the following page.

Review of manuscript : Hidden heatwaves and severe coral bleaching linked to mesoscale eddies and thermocline dynamics

Wyatt et al., submitted to Nature Communications

April 20, 2022

The authors use a combination of datasets – remotely sensed sea surface temperature (SST) and sea level anomaly (SLA), and in-situ measurements of temperature – to show that there could be a decoupling between SST and temperature at the depth of coral reefs that is linked to some mesoscale variability. They argue that sustained periods of enhanced temperature at depth that could qualify as marine heat waves (MHWs) cannot be detected from SST data only. The manuscript is well written and general concepts and results clearly articulated. My background is in physical oceanography and I do not have a fine knowledge of the literature on coral bleaching, so I must admit I am not qualified to judge how novel and significant the results are. However, I found the manuscript quite speculative and not quantitative. More importantly, I have an important concern about the methodology, which I believe impedes the publication of the manuscript in the present form. Some clarification is needed.

1 Major comments

1. **Methodology – Internal wave cooling, lines 613-627.** The authors apply the same methodology as in Wyatt et al. (2020), which I find very convoluted and perhaps controversial. Wyatt et al. (2020) claims that “Our approach was to compare each of the original temperature time series with a corresponding time series filtered and adjusted to estimate temperature conditions in the absence of internal waves.” and the processing to get the non internal wave (NIW) signal is as follows :

step 1 : Filter the raw signal into a high-frequency signal and a low-frequency signal, with a cutoff frequency equal to the inertial (Coriolis) frequency.

step 2 : Compute the root mean square of the high-frequency signal over a running window equal to the inertial period and add it to the low-frequency signal. The resulting signal is a first approximation of the NIW signal.

step 3 : Low-pass the first approximation of the NIW signal to get the final NIW signal.

Step 1 is a classic method to separate wave signals to non-wave signals (e.g., Nash et al., 2005). Step 2 is very questionable since it creates a significant shift of the NIW signal that corresponds to a warm bias, hence a cold bias in the internal wave signal. Indeed, Eq. (2) in the manuscript can be decomposed following the described data processing into :

Figure 1: Illustration of the filtering processing used in the manuscript on a temperature time series. (a) ‘Obs’ (black line) is the raw temperature and ‘LP’ (red line) is the low-pass signal. (b) ‘HP’ (blue line) is the high-pass signal and ‘rms of HP’ (cyan) is the running root-mean-square of the high-pass signal. (c) ‘NIW = LP + rms of HP’ (orange) is the non-internal wave signal and Obs and LP are repeated from (a). Here we clearly see the offset in NIW as compared to LP. (d) ‘IWC = NIW - Obs’ (black) is internal wave cooling as defined in the manuscript. Positive values are shaded blue and correspond to a cooling effect whereas negative values are shaded red and correspond to a warming effect.

$$\begin{aligned}
 \text{IWC} &= \text{NIW} - \text{Obs} \\
 \text{Obs} &= \text{NIW} - \text{IWC} \\
 \text{Obs} &= \underbrace{\text{LP} + \text{rms}(\text{HP})}_{\text{NIW}} + \underbrace{(\text{HP} - \text{rms}(\text{HP}))}_{\text{IWC}}
 \end{aligned}$$

This means that both wave and non-wave signals are artificially shifted by $\text{rms}(\text{HP})$, compared to traditional decomposition frameworks.

I repeated the authors’ processing on a temperature time series from a moored thermistor (Figure 1). Step 1 corresponds to panel (a) and step 2 corresponds to panels (b) and (c). One clearly sees a positive shift in NIW implied by adding the rms of the high-pass signal : orange vs red line in panel (c). As a result, IWC is shifted to positive values, i.e., stronger cooling effect (panel (d)).

Internal waves are freely oscillating perturbations around a background state. They are linear

in the general case and alternatively bring positive and negative anomalies to the background signal, with an average effect close to zero. I sincerely do not understand what could justify the addition of the rms of HP to LP and I think this needs to be thoroughly clarified and motivated.

2. **Presence of internal waves.** The links between mesoscale variability, thermocline dynamics, mixed-layer dynamics and internal waves are glossed over. A proxy for the potential of internal wave activity is stratification (e.g., Leaman and Sanford, 1975). The manuscript only qualitatively comments on how mesoscale eddies could affect stratification and internal wave activity. The authors could compute the vertical gradient of temperature (instead of the Brunt-Vaisala frequency, if they are missing salinity) to be more quantitative.

2 Minor comments

1. L23 : ‘internal-wave cooling’ is obscure and not defined in physical oceanography.
2. L65 : I reckon 12 weeks is approximately 3 months so I do not get the major difference between DHW and DHM. Please clarify.
3. L93 and 105 : Fig S3 is seemingly not the correct figure to refer to.
4. L127 : the choice of events is not deeply motivated and the reader could doubt about how these events could orient the results.
5. L150 : ‘has been shown to result’ is incorrect, the propagation of IW is by essence associated with isopycnal displacements.
6. L224 : ‘Oceanographic processes . . .’ is vague. Please clarify.
7. L233 : replace ‘changes’ with ‘variability’ ?
8. L277 : remove ‘embedded in large-scale currents’ ?
9. Figure 1 : panels (a) and (b) : color range could be changed to ≈ 27.5 -31 to better see the regional contrasts.
10. Figure 2 : background shading does not bring relevant information, it should be removed.
11. Figure 4 : Zoom in the top 120 m ? and show N^2 or $\partial T/\partial z$? What are the dotted lines ?

References

- Leaman, K. D. and Sanford, T. B. (1975). Vertical energy propagation of inertial waves: A vector spectral analysis of velocity profiles. *J. Geophys. Res.*, 80(15):1975–1978.
- Nash, J. D., Alford, M. H., and Kunze, E. (2005). Estimating internal wave energy fluxes in the ocean. *J. Atmos. Oceanic Technol.*, 22(10):1551–1570.
- Wyatt, A. S., Leichter, J. J., Toth, L. T., Miyajima, T., Aronson, R. B., and Nagata, T. (2020). Heat accumulation on coral reefs mitigated by internal waves. *Nature Geoscience*, 13(1):28–34.

Reviewer #2 (Remarks to the Author):

The manuscript is well-written and describes an interesting link between marine heat waves and the bleaching of corals in the north side of Moorea island. These marine heat waves are specifically linked to eddy dynamics, which are sampled first by satellite altimetric products but also by subsurface temperature measurements. These subsurface temperature measurements underscore a potential disagreement between surface and subsurface temperature, which highlight the need to sample subsurface temperature in order to define the presence of marine heat waves and coral bleaching. I found overall that the message of the manuscript is novel and robust and should interest communities in biology and physical oceanography. I had one main concern, detailed below, about the variables used to detect internal waves.

This study carefully compares 2 events in 2016 and 2019. At first, these events have similar surface temperatures but their effects on coral bleaching are variable. The event in 2016 is associated with less heating and more cooling hypothesised to come partly from internal waves cooling. In contrast, the event in 2019 has a strong warm structure across the water column, which is associated with less internal waves. The authors convincingly explain the effect of eddies on subsurface temperatures. I found the link with internal waves harder to follow mostly because of the "unconventional" definitions used to categorise internal waves. The authors defined internal waves by high and low-pass filtering temperature signals in 2 variables but also by using additional processing briefly mentioned from a previous study (Wyatt et al., 2020): Non-Internal Wave NIW and Internal Wave Cooling IWC p.23. Because of the difficulty to understand what NIW (and subsequently IWC) truly represent, I could not fully grasp the discussion of Fig.2, Fig.3, and Fig.S2.

Thus, the link between internal waves and eddies/marine heat waves is unclear. Although I understand that the main message/discussion is around the effect of eddies on marine heat waves, the authors should better explain the internal wave signal and specifically the paragraph starting l.159 and the one starting l.613. More guidance is provided below.

I also have a long list of minor comments that should help to clarify the text. The manuscript was hard to follow because of these errors and because of multiple errors in referencing the figures.

My expertise is mainly in physical oceanography so I have some understanding of eddy and internal waves dynamics. However, I could not provide much comments on the ecological side of the manuscript, although, most arguments seem rather convincing.

Main comment:

I found Fig.2 and its caption rather misleading as you are introducing the caption with "low- and high-frequency temperature variations" (l.801) and mentioning "relative high-frequency water temperature variations" (l.802). Furthermore, the text indicates that you are actually doing a low-pass and a high-pass filter (l.159, 161, 163), which are then displayed in Fig.2 but this is not the case, You should describe something simpler in this caption such as the low-frequency variability, which I think is NIW, and then the observed temperature, which I think is the black line. Thus, something more or less like the caption of Fig.2 in Wyatt et al. (2020). Then in the text you can mention that you have high-frequency variability in the observed signal in the paragraph starting l.159, when you refer to Extended Data Fig.2.

Otherwise, if your black line in Fig.2 is not the observed temperature, you should be more explicit about its definition in the caption and in the text. For example, by reading l.171-173, I tend to think that your black line could be related to IWC.

Also, I don't think that NIW is well defined in the manuscript. Its definition is probably

better explained in Wyatt et al. (2020) but I still find it difficult to grasp exactly what NIW means. You should expand its definition around l.615 and probably add a simple plot that would explain the differences between NIW and a simpler low-pass filtered timeseries. I suspect that by using a simple low-pass filter signal you would still have the same message in Fig.2. The only difference would then be that your white lines (for NIW) would be lower, so you would probably need a different 'bleaching threshold' than MMM+1degC. By the way, is this bleaching threshold common in the literature or could it be lowered, for example simply to MMM?

Minor comments:

- L.21-23: I find this sentence unclear and the links between the clauses are difficult to follow. Similarly, I don't understand what "local ocean weather" means.
- L.36: 'working definitions' is vague, replace.
- L.40 and l.50: you use "in-situ" and "in situ", try to be consistent in the full manuscript.
- Fig.1a-b: Could you please add the name of Moorea and the other islands (Tahiti...) at least on this first figure?
- L.55: "confidence intervals" are loosely defined, confidence intervals of what exactly?
- L.105: Fig.S3e,f; should it be Fig.S3d,f instead?
- L.108: Refer to Table.S2 after (2.10degC-days). I guess the numbers given l.103 are not reported in the tables/figures, they come from reference 35, is that correct? Is this reference using a 2 by 2 degrees box?
- Fig.S1a: The blue and green lines are hard to distinguish, chose another color for one of them. Then, your lines are all hidden behind the red one. Maybe use different linewidths or something else to clearly distinguish these lines.
- L.135: add a reference to Fig.1a,b after (30.2degC).
- L.143: Is p the correlation coefficient?
- L.145-146: This statement contradicts the previous sentence where the correlation was higher at 40m than 10m.
- L.147: Do you mean "attenuation" instead of "amelioration"?
- L.180: "similar" is quite strong when comparing Fig.2p and Fig.2l.
- L.194: replace "which" by "with"?
- L.197: add "between" at the beginning of the line.
- L.206: Should it be Fig.2g-h instead of Fig.2c-d?
- L.237: Fig.5 seems to be mentioned before Fig.4.
- L.256: Fig.4b should be Fig.4a.
- L.258: Fig.4c should be Fig.4b for Mar-Apr period.
- L.269: Replace "depths < 50 m" by "depths shallower than 50 m" and similarly for other occurrences. Inequality signs are confusing specifically when you're using positive depths in the text but negative depths in the figures (Fig.1,..).
- L.269: replace "surface mixed layer" by "mixed layer depth", same l.273.
- L.274: "...thermocline is more...".
- L.316: Cut the sentence, the link is unclear between the 2 clauses.
- L.580: The temperature recorders at 4 different depths are at different locations. This was unclear from only reading the text and figures. You could add a subplot in Fig.S1 by zooming north of Moorea and showing the locations of these recorders. Alternatively, you could add the latitude/longitude of each instrument in Table S1.
- l.616: I don't understand the meaning of "to isolate removed frequencies".
- L.625: You missed a word after also, maybe add "calculated".
- It's not clear from the paragraph l.613 why you would calculate IWC with eq.2 instead of simply applying a high-pass filter.
- L.631: Is n the number of photoquadrats, clarify and maybe change notation as n is already used l.596?
- L.642: How do you define "offshore from Moorea" exactly? Are you using a box around Moorea or one of the boxes in Fig.S1d? What is the depth associated with this box given that Argo floats are mostly in deep water? Specifying this should help to understand l.252 and l.816. For example, l.252 refers to deep water below 1500 m whereas Fig.4a stops at 200 m deep.

- **Table 1:** Try to be consistent by reporting numbers with two decimals everywhere, see for example column 2 and 3.
- **L.806:** "shown for reference".
- **Fig.5b:** Use a different color than the dashed grey line, it's hard to see behind the black curve.
- **L.834:** b-c should be c-d.
- **L.894-896:** Unclear and long sentence.
- **Extended data Fig.2:** I can't really distinguish the black/blue lines/contours, maybe use another color. The fontsize of the figure is too small. Maybe do a plot in 2 by 2 subplots, also expand the x-axis width, this could probably help to remove the tiny insets with illegible captions.
- **L.897:** MHWs. "the most severe events".
- **L.898:** Fig.S1a-b is about DHD not SST.
- **L.900:** I don't understand where "17 days" comes from, Tab.S2 would say 14 days.
- **L.906:** Provide a reference to the Nino 3.4.
- **L.906:** Should the second comma be after "the event"?
- **L.909:** not 2 but 3 MHWs not evident at a regional scale, what about April 1991?
- **L.910:** (Fig.S1a; Table S3 vs Table S2).
- **L.911:** "detection" instead of "localization".
- **L.939:** Should it be Fig.S1a instead of Fig.S1b?
- **L.945:** Replace "interrogate".
- **L.947:** Should it be Fig.S1a instead of Fig.S1b?
- **L.982:** What is pseudo-F and p?

Reviewer #3 (Remarks to the Author):

Overall, I think that this is a compelling paper that emphasizes the need to understand the local and regional marine weather that can affect the vertical distribution of heat on coral reefs. It builds on our understanding of the role of internal waves in mitigating MHW, and introduces the role of the regional eddy field in either emphasizing or quelling the potential for IWC to mitigate MHWs.

I am not aware of other data sets that include such a long history of in situ temperature records, although I suspect that they should be available from the GBR through the efforts of AIMS. But I have not seen an analysis of the role of eddies in this context. The length of the record and the associated open ocean profile data provide a unique perspective these processes.

One note that I would make is that ICW may also contribute the flux of a planktonic food supply. Unfortunately, the measurements for this are harder to come by, although more recent technology may facilitate those types of measurements. The damping of the ICW may damp the supply of a planktonic food supply to the corals which could reduce the ability of the coral to adapt to the effects of the MHW.

Specific comments:

1. Line 219 – perhaps the majority of reefs do experience "some degree of IWC". But there are regions where IWC is a rare occurrence.
2. Line 224 – Some of the "processes seaward of reef slopes" may themselves bring cooling to the reef. Anticyclonic eddies have uplifted isopycnals along their periphery. Depending on the water column structure these in themselves may alter the temperature environment on the reef. This is opposite in effect to what is described in this paper. It's not contradictory, but rather depends on how the oceanographic feature intersects with the shallow structure of the reef.
3. Line 283 – the "SLAs" are propagating. Should we deduce that the eddies are westward propagating as well?
4. Line 344 "centred" – Is American or UK English being used?
5. Line 352 – when "nutrient upwelling" is mentioned, is that inorganic nutrients, or

planktonic food supply? I suspect with the mention of bleaching that it refers to inorganic nutrients. An interesting question is whether with the suppression of internal waves, planktonic nutrition to the reef is also suppressed, perhaps exacerbating the effects of the MHW.

#	Comment	Response
1.1	The authors use a combination of datasets – remotely sensed sea surface temperature (SST) and sea level anomaly (SLA), and in-situ measurements of temperature – to show that there could be a decoupling between SST and temperature at the depth of coral reefs that is linked to some mesoscale variability [...] The manuscript is well written and general concepts and results clearly articulated.	Thank you for your positive comments and help further improving the manuscript. As also recognised by the other reviewers, our central finding is novel, robust and compelling: depression of the thermocline and internal-wave impacts associated with mesoscale eddy propagation provides a mechanistic explanation of unanticipated heat accumulation and ecologically catastrophic coral bleaching that appears paradoxical based on regional surface conditions.
1.2	My background is in physical oceanography and I do not have a fine knowledge of the literature on coral bleaching, so I must admit I am not qualified to judge how novel and significant the results are. However, I found the manuscript quite speculative and not quantitative.	Thank you for providing an important perspective on the physical oceanographic aspects of our work, and for comments on areas that may have appeared more speculative. One of the great challenges of this kind of analysis is to marry long-term oceanographic data with reef-level ecological observations over a similar timeframe. While we would not entirely agree that the links between the observed physical and biological time series are speculative, it is true that we are looking at associational patterns between co-varying physical and biological events, and of course this kind of real-world environmental analysis includes organismal responses, so we cannot represent the findings as controlled experiments. However, the biological responses we report do show that the 2019 coral bleaching event was ecologically catastrophic, with high rates of subsequent mortality and a reshaping of the coral community not evident in at least a decade (see Fig 6e and Burgess et al. (2021)). Surprisingly, the thermal signature of this event was not evident based on sea surface temperature (SST) metrics commonly relied on by ecologists for marine heatwave assessments. By contrast, the 2016 event was evident in SST metrics, but widespread coral bleaching was absent. Our unravelling of the paradoxical contrasts between 2019 and 2016 serves to demonstrate a link between broader eddy and internal wave dynamics, and consequently ecologically significant coral bleaching events. Our novel observations show that eddy dynamics influence internal-wave cooling on a coral reef ecosystem and can thus alter thermal stresses experienced during heatwaves. The analyses also highlight the benefits of considering broader-scale physical oceanography to better understand reef-level thermal environments and the responses of corals to marine heatwaves. We have added further quantitative evidence of these links as below (#1.8). We also note that Reviewer 2 found the results were “novel and robust and should interest communities in biology and physical oceanography” and Reviewer 3, coming from an ecological background, described it as a “compelling paper that emphasizes the need to understand the local and regional marine weather that can affect the vertical distribution of heat on coral reefs”. Based on the comments from all three reviewers we believe that the revised manuscript can provide an interesting and novel demonstration that high-resolution oceanographic observations coupled with remotely sensed and in-situ heating metrics are essential to elucidate the drivers of marine heatwaves. We hope that publication of the paper in a high-profile journal will help to motivate a step-change improvement in linking oceanographic processes more appropriately in studies on the environmental forcing of coral bleaching.

#	Comment	Response
1.3	More importantly, I have an important concern about the methodology, which I believe impedes the publication of the manuscript in the present form. Some clarification is needed.	As detailed further below, the methodology for isolating the signal of internal waves in reef temperature records taken at reef-level in shallow water (e.g., depths less than 40 m) does appear to be robust and has been tested on multiple time series (Wyatt et al., 2020). However, we understand that clarification was needed, especially relative to consideration of internal waves in an open ocean context versus a steep reef slope. The interaction of internal waves with topography such as a coral reef slope is a very different case to that of the open ocean. A key point we may have failed to adequately present in the earlier version of manuscript is that the method applied here works in the particular case of internal-wave thermal variations at reef-level where the sensors are located above the mean thermocline depth, and that internal waves are characterised by rapid downward temperature deviations. We provide further clarification below and have thoroughly revised the text accordingly. We do also note here that our technique for filtering the internal wave signals in reef-level time series is not a central point of the current manuscript, but rather allows us to apply a consistent method across different years as part of investigating the subsurface dynamics contributing to the catastrophic coral bleaching in 2019. We are thereby able to demonstrate that the ecological impact of this latest heatwave was associated with a period of reduced internal-wave cooling in 2019 as eddies depressed the thermocline. Conversely, during the 2016 heatwave persistent internal-wave cooling likely prevented coral bleaching, even during what appeared to be a significant marine heatwave at the sea surface.
1.4	The authors apply the same methodology as in Wyatt et al. (2020), which I find very convoluted and perhaps controversial.	Clarification regarding the previously published internal wave filtering approach is provided in the following comments and revised text as requested. We do note that Wyatt et al. (2020) was evaluated through two rounds of careful peer review and published in Nature Geosciences; it has already been cited frequently in the short time since publication.
1.5	Step 1 is a classic method to separate wave signals to non-wave signals (e.g., Nash et al., 2005). Step 2 is very questionable since it creates a significant shift of the NIW signal that corresponds to a warm bias, hence a cold bias in the internal wave signal. [...] This means that both wave and non-wave signals are artificially shifted by rms(HP), compared to traditional decomposition frameworks.	In “Step 1” we do, indeed, take a standard approach of frequency filtering to isolate variability in the temperature time series corresponding to the dominant semi-diurnal frequency band and make an estimate of the low-frequency temperature signal. A key point, however, is that we are analysing the temperature time series from bottom-mounted temperature sensors at shallow depths above the mean depth of the thermocline. Thus, the observed deviations in temperature within the semi-diurnal frequency band correspond almost entirely to downward spikes in temperature associated with the upward displacement of cool water isotherms onto the reef. This is a very different case than analysing, e.g., temperature time series from instruments on a vertical mooring in deep water and spanning the mean thermocline depth. In our case, the goal was not to achieve a standard frequency filtering outcome, but rather to approximate what the low-pass component of the temperature signal would have been had there not been imbedded high-frequency downward temperature deviations. In developing the method described in Wyatt et al. (2020), which involved testing the results against multiple reef-level time series with and without internal waves present, we found by visual and empirical analysis that the downward deviations in temperature over these shallow reef habitats (above the mean thermocline depth) are well approximated by the RMS variability within the semi-diurnal frequency band: RMS(HP). Therefore, after initially filtering to isolate the high- and low-pass components of the observed temperature time series, we then (“Step 2”) must add to the low-frequency signal a component corresponding to the moving window RMS value of the high-frequency signal. This yields our best estimate of the temperature signal that would have been observed had the downward spikes in temperature from internal waves not been present. We term this new, estimated signal the “Non-Internal Wave” (NIW) time series. Again, visual and empirical analysis of time

#	Comment	Response
		series during times with and without internal waves indicated we produced a close approximation of the time series absent the high-frequency cooling events (see the observed versus NIW signals after May in 2016 and 2019 in Fig 2a-h). A detailed justification for the need to shift the low-pass filtered data to accurately represent non internal wave conditions was provided in Wyatt et al. (2020) and is supported by examination of data in that paper and the present manuscript (see also #1.7). As reported there the verbatim explanation was: Visual examination of the temperature time series [at sites across the Pacific Ocean], and of conditions at the sites during internal wave events [...] showed that the effect of internal waves was typically characterized not by temperature fluctuations that are symmetric around the daily running mean, but by repeated bouts of rapid cooling and subsequent return to warmer, ambient conditions [...] Such temperature fluctuations represent upward incursions of subsurface water onto the reefs, forced by internal waves [as supported by reef studies on the Great Barrier Reef, French Polynesia (the present site) and Florida Keys, see (Wyatt et al., 2020) for refs] Therefore, a simple low-pass filtering of the data would not have accurately represented conditions had internal waves not been present at a given site. Instead, to account for the effect of variability associated with internal waves, the approach was to adjust the low-pass, low-frequency signal for each depth and site by an amount corresponding to a moving window root mean square (r.m.s.) value of the high-frequency variability. Although we recognise that other approaches could be used to derive a hypothetical “non-internal-wave” time series, there is no perfect solution for estimating the thermal environment without internal waves from time series that contain their influence. None-the-less, there are several lines of evidence that the filtering approach is sufficiently robust for estimating what reef-level thermal environments would be without internal waves. As detailed in Wyatt et al. (2020), our filtering closely matches the observed timeseries during periods when internal waves are absent due to: (1) deepening stratification or strong mixing, such as during winter (see white lines in Fig 2 of Wyatt et al. (2020), and the end of the timeseries in present Fig 2a-h) or (2) during tropical storms (analysis in preparation). We can now add a new case based on the present analysis, eddy-induced thermocline depression, such as during 2019: compare the observed (black) and filtered (white) signal in Fig 2m-p. The fact that the interaction of internal waves with the reef slope does not create a symmetrical oscillation about a mean state is evident through such visual examination of the data, especially when the internal waves were weaker, and less breaking and mixing likely occurred, during 2019 (see especially Fig 2p). It would not be accurate to represent a non-internal wave signal as a simple low pass filter of the observed data, since the internal waves clearly represent bouts of rapid cooling below a mean state (i.e., not linear oscillations about a mean). For clarity, we should again emphasise that the internal wave filtering approach described in Wyatt et al. (2020) is not central to the main finding of the present paper: that eddy-induced thermocline depression reduced the OBSERVED internal wave field. We believe this main finding would stand regardless of the filtering approach taken to estimate a non-internal-wave signal. In fact, in addition to those discussed in Wyatt et al. (2020), the 2019 event when internal waves were reduced provides another strong demonstration that the filtering approach robustly creates internal-wave-free time series from reef-level observations.
1.6	I repeated the authors' processing on a temperature time series from a moored thermistor (Figure	As above, while we understand the reviewer's concern, we respectfully point out that application of the method we describe to the more general case of thermistor records on a vertical mooring is a different situation than bottom-mounted thermistor time series in shallow water. Indeed, if we apply the method in Wyatt et al. (2020) to time series where there are symmetrical or quasi-symmetrical high-frequency temperature variations above and below a

#	Comment	Response
	1). Step 1 corresponds to panel (a) and step 2 corresponds to panels (b) and (c). One clearly sees a positive shift in NIW implied by adding the rms of the high-pass signal : orange vs red line in panel (c). As a result, IWC is shifted to positive values, i.e., stronger cooling effect (panel (d)).	low-frequency (e.g., running mean) signal, then “Step 2” of adding the RMS of the high-frequency signal could introduce a warm bias in the resulting estimated signal. However, again, we do not believe that this moored thermistor data analysed by the reviewer is relevant to the question at hand. Those data, presumably measured at single point in the open / deep ocean using a mooring, reflects the behaviour of internal waves in the absence of interaction with topography. It is very important to emphasise that our data are measured at reef level, with instruments directly bottom-mounted on the coral reef slope and thus capture the temperature variations induced over corals as the internal waves break and run-up the coral reef slope. The stark difference between internal waves measured from an open ocean mooring to those measured at reef-level may be analogous to the difference between surface gravity waves measured offshore in deep water versus when they break/run up a beach. We have highlighted in the text that reef-level measurements are taken at each depth over the reef slope (L630-635, new Fig S1e), rather than at a single point in space such as with a mooring, and thus reflect the interaction of internal waves with the reef topography, which cannot be captured with point mooring measurements in the open ocean. We further note that evidence of internal waves in the reviewer’s data appears negligible in the context of our tropical ocean observations, with the data appearing to be from a high latitude region (temperatures of ~ 7 °C) where stratification depth was likely shallow. This is unlikely to be representative of the internal wave processes we have measured associated with much deeper tropical stratification (e.g., ~30 to 50 m surface isothermal depth and maximum thermocline stratification at ~60 m in summer around Moorea (Leichter et al., 2012)), and, again, the reviewer’s test data do not seem to represent the interaction of internal waves with a steep reef slope where the sensors are located above the mean thermocline depth.
1.7	Internal waves are freely oscillating perturbations around a background state. They are linear in the general case and alternatively bring positive and negative anomalies to the background signal, with an average effect close to zero. I sincerely do not understand what could justify the addition of the rms of HP to LP and I think this needs to be thoroughly clarified and motivated.	We appreciate the clarity provided by this comment, and agree that clarification was needed here in addition to comments above. We thank the reviewer for making clear the importance of emphasising the particular situation in our study and we now do that more explicitly in the text. We have revised the description of the filtering approach (L666-692) and added a new supplementary figure for visualisation (Fig S2) to reflect the following points. In the case of the run-up of breaking internal waves on a coral reef (e.g., Leichter et al., 1996; Davis et al., 2008; Leichter et al., 2012) or in other more beach-like settings (e.g., Pineda, 1991; Pineda & Lopez, 2002; Scotti et al., 2008) in shallow water the observed internal waves are in fact highly non-linear, and resulting temperature signals are not characterized by alternating positive and negative anomalies around a freely oscillating background state. This can be the case for open ocean (deep water) observations, such as the moored thermistor data the reviewer uses as an example, but is not likely to be the case for internal waves propagating, and often breaking, on a reef slope. Again, as described above, a very important distinction to make here is that our data are measured at reef level, bottom-mounted on the coral reef slope and above the typical thermocline depth and thus reflect the interaction of the internal waves pushing deep water up and onto the reef – our data clearly show that, at the reef depths studied, this manifests as periodic drops in temperature as the internal waves run up / break on the reef slope.
1.8	The links between mesoscale variability, thermocline dynamics, mixed-layer dynamics and internal waves are glossed over. A proxy for the potential of internal wave activity is stratification (e.g., Leaman	In response to the reviewer’s comment, we have conducted a new analysis and added a figure (Extended Data Fig 4) that better shows quantitative links between the eddy-induced sea-level variation and stratification / mixed-layer depths to the extent allowed by available ARGO profiling data. Fortunately, we were able to identify ARGO profiles near Moorea that captured the peak of the 2019 MHW and warm-core eddy-induced sea-level elevation. ARGO profiles during 2016 occurred prior (9 Feb – 1 Mar) to the peak of the MHW and sea-level depression (Apr), so we expect underestimated the stronger or shallower

#	Comment	Response
	and Sanford, 1975). The manuscript only qualitatively comments on how mesoscale eddies could affect stratification and internal wave activity. The authors could compute the vertical gradient of temperature (instead of the Brunt-Vaisala frequency, if they are missing salinity) to be more quantitative.	stratification that led to strong internal-wave forcing; nonetheless the 2016 profiles also provide a useful comparison under more moderate SLA and shallower stratification to the 2019 event when there was a clear effect of mesoscale eddies on stratification and mixed-layer depths: Stratification was also weaker and deeper in ARGO profiles north of Moorea during 2019 (Extended Data Figure 4a), with the depth of the maximum buoyancy frequency (N) on average 40 m deeper during 2019 (99.9 ± 51 m) compared to 2016 (62.0 ± 9.9 m; Extended Data Figure 4b). (L277-280) ...SLA elevations offshore were associated with deeper maximum N in 2019 profiles (Extended Data Figure 4b). (L282-283) We agree that broader analysis of the detailed mechanistic links between mesoscale eddies and stratification could be of considerable interest. However, this would take a concerted physical oceanographic study across large spatial and temporal scales and is beyond the present scope of our study or available data. While the new analysis we have added is useful for providing further quantitative evidence of the links between eddies, stratification and internal waves, and we thank the reviewer for raising it, we should note that in a way we had already quantified water column dynamics from a viewpoint of greater relevance to reef-level temperatures and bleaching: the marked internal wave dynamics we quantified over the reef could reflect reduced stratification, deepened mixed layer depths, or both, but the effects on reef-level temperature patterns remains the same from the perspective of coral organisms and reef ecology.
1.9	L23 : 'internal-wave cooling' is obscure and not defined in physical oceanography.	We have clarified that this is a novel term (L666) that we have defined to describe the cooling of coral reef (or other coastal habitats) due to the interaction of internal waves with coastal topography.
1.10	L65 : I reckon 12 weeks is approximately 3 months so I do not get the major difference between DHW and DHM. Please clarify.	DHW uses 12 weeks of weekly SST data ($n = 12$) while DHM uses 3 months of monthly SST data ($n = 3$). The two metrics represent a progression in precision as the temporal resolution of satellite estimates of SST increased from monthly to weekly. The latest satellite products now provide SST at daily resolution and this higher resolution is the focus of our analyses. As we demonstrate, significant information on the progression of marine heatwaves can be lost using low resolution data, or data down-sampled or averaged to, e.g., weekly (see Fig S3 for a comparison of weekly / daily, DHW / DHD metrics)
1.11	L93 and 105 : Fig S3 is seemingly not the correct figure to refer to.	Fig S3 (now Fig S4) provided a comparison of the heatwaves computed as DHW (Fig S4e-f) or DHD (Fig S4g-h), to highlight the loss of resolution provided by DHW calculations. We have confirmed the figure reference is correct on both previous lines L93 and L105.
1.12	L127 : the choice of events is not deeply motivated and the reader could doubt about how these events could orient the results.	The events were selected based on observations of heatwave conditions above the regional coral bleaching threshold (as degree heating days above 1 °C-days). We now clarify in the text that the events were selected quantitatively based on this observed heat accumulation (L664-665).
1.13	L150 : 'has been shown to result' is incorrect, the propagation of IW is by essence associated with isopycnal displacements.	This has been adjusted to "has been shown to be associated with" (L158)

#	Comment	Response
1.14	L224 : 'Oceanographic processes . . . ' is vague. Please clarify.	Specific processes were provided in the proceeding sentence; this has now been clarified with "Such oceanographic processes" (L243).
1.15	L233 : replace 'changes' with 'variability' ?	Done.
1.16	L277 : remove 'embedded in large-scale currents' ?	Changed to "associated with regional current systems" (L305)
1.17	Figure 1 : panels (a) and (b) : color range could be changed to 27.5-31 to better see the regional contrasts.	We understand the reviewer's thinking here, however we chose the regional colour scale to match the in-situ temperature scale in the panels below for comparability between the SST and subsurface thermal conditions. Clearer regional contrast for the SST data can be found in Extended Data Figure 1.
1.18	Figure 2 : background shading does not bring relevant information, it should be removed.	We respectfully disagree with the reviewer here – the background shading provides the daily SST temperature and is helpful in placing the in-situ temperatures in the context of SST, especially to highlight that they are regularly significantly cooler (in the blue shaded region) than daily SST due to internal-wave cooling.
1.19	Figure 4 : Zoom in the top 120 m ? and show N2 or @T=@z ? What are the dotted lines ?	The selection of 200 m was arbitrary, but is often considered as the surface layer – selection of 120 m instead would cut off the 26 deg isotherm deepening during the 2019 event, which we refer to in the text. As detailed in #1.8, we have added a new analysis and figure showing the vertical distribution of buoyancy frequency (N) based on ARGO profiles in 2016 and 2019 and links to SLA (Extended Data Fig 4). As per the caption: Dotted lines in (b-c) show average satellite SLA north of Moorea (see Fig. 5) for comparison to the reef-level SLA (solid lines)

References (Responses to Reviewer #1)

- Burgess, S.C., Johnston, E.C., Wyatt, A.S.J., Leichter, J.J. and Edmunds, P.J. (2021) Response diversity in corals: hidden differences in bleaching mortality among cryptic Pocillopora species. *Ecology* **102**: e03324.
- Davis, K.A., Leichter, J.J., Hench, J.L. and Monismith, S.G. (2008) Effects of western boundary current dynamics on the internal wave field of the Southeast Florida shelf. *Journal of Geophysical Research* **113**.
- Leichter, J.J., Stokes, M.D., Hench, J.L., Witting, J. and Washburn, L. (2012) The island-scale internal wave climate of Moorea, French Polynesia. *Journal of Geophysical Research-Oceans* **117**.
- Leichter, J.J., Wing, S.R., Miller, S.L. and Denny, M.W. (1996) Pulsed delivery of subthermocline water to Conch Reef (Florida Keys) by internal tidal bores. *Limnology & Oceanography* **41**: 1490-1501.
- Pineda, J. (1991) Predictable upwelling and the shoreward transport of planktonic larvae by internal tidal bores. *Science* **253**: 548-551.
- Pineda, J. and Lopez, M. (2002) Temperature, stratification and barnacle larval settlement in two Californian sites. *Continental Shelf Research* **22**: 1183-1198.
- Scotti, A., Beardsley, R.C., Butman, B. and Pineda, J. (2008) Shoaling of nonlinear internal waves in Massachusetts Bay. *Journal of Geophysical Research: Oceans* **113**.
- Wyatt, A.S.J., Leichter, J.J., Toth, L.T., Miyajima, T., Aronson, R.B. and Nagata, T. (2020) Heat accumulation on coral reefs mitigated by internal waves. *Nature Geoscience* **13**: 28-34.

#	Comment	Response
2.1	The manuscript is well-written and describes an interesting link between marine heat waves and the bleaching of corals in the north side of Moorea island. These marine heat waves are specifically linked to eddy dynamics, which are sampled first by satellite altimetric products but also by subsurface temperature measurements. These subsurface temperature measurements underscore a potential disagreement between surface and subsurface temperature, which highlight the need to sample subsurface temperature in order to define the presence of marine heat waves and coral bleaching. I found overall that the message of the manuscript is novel and robust and should interest communities in biology and physical oceanography [...] This study carefully compares 2 events in 2016 and 2019 [...] The authors convincingly explain the effect of eddies on subsurface temperatures.	Thank you for your positive comment on the novel and robust nature of the findings.
2.2	I found the link with internal waves harder to follow mostly because of the “unconventional” definitions used to categorised internal waves. The authors defined internal waves by high and low-pass filtering temperature signals in 2 variables but also by using additional processing briefly mentioned from a previous study (Wyatt et al., 2020): Non-Internal Wave NIW and Internal Wave Cooling IWC p.23. Because of the difficulty to understand what NIW (and subsequently IWC) truly represent, I could not fully grasp the discussion of Fig.2, Fig.3, and Fig.S2. Thus, the link between internal waves and eddies/marine heat waves is unclear. Although I understand that the main message/discussion is around the effect of eddies on marine heat waves, the authors should better explain the internal wave signal and specifically the paragraph starting l.159 and the one starting l.613. More guidance is provided below.	We understand the need to clarify this point in the text, since the reviewer’s description of the method for quantification of internal waves suggests some misunderstandings or lack of clarity on our part. As described above for Reviewer 1 (#1.3-1.7), we now provide more detail to clarify the multiple steps in the analysis that produces an estimate of the temperature signals that would have been observed in the absence of internal waves: our “Non-Internal-Wave” (NIW) signal. We should again note as above for Reviewer 1’s comment (#1.5) on the internal wave filtering, that the internal wave filtering approach described in Wyatt et al. (2020) is not central to the main finding of the present manuscript, but does provide a consistent means of comparison between the subsurface heatwaves at Moorea, especially in terms of placing the ecologically catastrophic 2019 event into a longer temporal context. The main observation is eddy-induced thermocline depression in 2019 reduced the OBSERVED internal wave impacts on the reef and thus overall cooling, leading to hot subsurface temperatures and severe bleaching. The reviewer also notes that the filtering part of the analysis is not central to the main findings of the manuscript. We have altered the text regarding the quantification of internal waves (L666-692), including in relation to the reviewer’s specific comments below, and thank the reviewers for their help in bringing the need for clarification to our attention.
2.3	My expertise is mainly in physical oceanography so I have some understanding of eddy and internal waves dynamics. However, I could not provide much comments on the ecological side of the manuscript, although, most arguments seem rather convincing.	Thank you for your physical oceanography perspective on this work. As noted for Reviewer 1 (#1.2), this perspective is highly valuable for improving our analyses and presentation. Fortunately, the ecological aspects of the study have also been found to be compelling by Reviewer 3, who has an ecological background.

#	Comment	Response
2.4	I found Fig.2 and its caption rather misleading as you are introducing the caption with “low- and high-frequency temperature variations” (l.801) and mentioning “relative high-frequency water temperature variations” (l.802). Furthermore, the text indicates that you are actually doing a low-pass and a high-pass filter (l.159, 161, 163), which are then displayed in Fig.2 but this is not the case, You should describe something simpler in this caption such as the low-frequency variability, which I think is NIW, and then the observed temperature, which I think is the black line. Thus, something more or less like the caption of Fig.2 in Wyatt et al. (2020). Then in the text you can mention that you have high-frequency variability in the observed signal in the paragraph starting l.159, when you refer to Extended Data Fig.2. Otherwise, if your black line in Fig.2 is not the observed temperature, you should be more explicit about its definition in the caption and in the text. For example, by reading l.171-173, I tend to think that your black line could be related to IWC.	Thank you for suggestions on improving this caption. The black line in the figure is the observed temperatures. As suggested, we have simplified the opening to “Contrasting observed and non-internal-wave temperature variations in reef-level temperatures across Moorea’s north shore during the 2016 and 2019 MHWs” (L877-878). We have focused on observed and non-internal-wave in the rest of the caption (rather than high and low frequency) to avoid confusion.
2.5	I don’t think that NIW is well defined in the manuscript. Its definition is probably better explained in Wyatt et al. (2020) but I still find it difficult to grasp exactly what NIW means. You should expand its definition around l.615 and probably add a simple plot that would explain the differences between NIW and a simpler low-pass filtered timeseries. I suspect that by using a simple low-pass filter signal you would still have the same message in Fig.2.	As described above, we do appreciate these comments, especially since we are close to the analysis and methods and probably understand the specificity of the data and application better than we had described them for readers. We agree that a simple low pass filter would still tell a compelling story, but believe that our approach more accurately describes a non-internal-wave signal over a reef slope, where temperatures are subject to sudden drops (rather than oscillations around a mean) as internal waves propagate or break up the reef slope. See details provided above (#1.5-1.7).
2.6	The only difference would then be that your white lines (for NIW) would be lower, so you would probably need a different ‘bleaching threshold’ than MMM+1degC. By the way, is this bleaching threshold common in the literature or could it be lowered, for example simply to MMM?	There is no connection between the bleaching threshold and the non-internal-wave signal. As per convention in the coral bleaching literature, the bleaching threshold in coral reef studies is set at the “maximum monthly mean” plus 1 °C (MMM+ 1 °C) sea surface temperature (SST) for a region. While we likely agree with the reviewer that this threshold is somewhat arbitrary, we believe it is best to use this frequently applied approach to remain consistent with the relevant coral bleaching literature. Although the use of SST-based thresholds is widespread throughout the coral bleaching literature it may, indeed, underestimate depth-specific physiological stresses. Importantly here however, we note that applying a bleaching threshold based on SST across depths as we have done (1) leads to consistent, comparable heat metrics across depths, and (2) provides conservative estimates of the ecologically relevant heating with increasing depth. As detailed in Wyatt et al. (2020), in-situ MMMs would decrease with depth relative to surface-based MMMs, especially if high-resolution internal-wave cooling is considered. Therefore, the predicted

#	Comment	Response
		heat stress associated with a MHW based on a surface threshold could underestimate the true physiological stress at depth. On the other hand, calculating depth-specific degree-heating based on in-situ MMMs would hamper quantitative comparisons of MHW-associated heating across depths. This point is made in the manuscript: The DHD estimates are conservative since SST-derived MMM are likely higher than in-situ MMMs, e.g., by up to 2.7 °C relative to in-situ temperatures at 40 m at Moorea (Wyatt et al., 2020). (L652-653) There is also little evidence in the literature of depth-based differences in bleaching responses to similar temperature stresses. This is partly a reflection that coral physiology and bleaching studies are rarely conducted across a broad range of depths. The question as to whether corals in cooler/deeper habitats have a different bleaching threshold to conspecifics in hotter/shallower is not resolved, with some evidence that bleaching responses could be relatively intrinsic rather than environmentally driven (Barott et al., 2021). We reiterate here, that the details of the MMMs and DHD are not in any way central points of our contribution, but simply standard approaches that are well developed and widely known in the coral bleaching literature.
2.7	L.21-23: I find this sentence unclear and the links between the clauses are difficult to follow. Similarly, I don't understand what "local ocean weather" means.	This sentence has been revised to make it easier to follow (L21-24). 'Ocean weather' is a concept introduced in the referred paper in Nature by Bates et al. (2018), which implores marine ecologists to consider variations in oceanographic conditions in much the same way that terrestrial ecologists consider weather variations on land. We do not have space in the abstract to expand on the detail provided in that reference.
2.8	L.36: 'working definitions' is vague, replace.	We have deleted "working" (L39).
2.9	L.40 and L.50: you use "in-situ" and "in situ", try to be consistent in the full manuscript.	We have attempted to consistently hyphenate in situ only when it is a compound modifier.
2.10	Fig.1a-b: Could you please add the name of Moorea and the other islands (Tahiti...) at least on this first figure?	Labels have been added in Fig 1a-b for Moorea and Tahiti (L861).
2.11	L.55: "confidence intervals" are loosely defined, confidence intervals of what exactly?	This is explicitly defined in the following words: "calculated seasonally based on historical SST" (L59).
2.12	L.105: Fig.S3e,f; should it be Fig.S3d,f instead?	This is intended to highlight the delay in the peak of DHW metrics after the two MHW in 2016 and 2019, which are in panels e and f in Fig S3.
2.13	L.108: Refer to Table.S2 after (2.10degC-days). I guess the numbers given L.103 are not reported in the tables/figures, they come from reference	Reference to Table S2 added. That is correct, the values that were on line 103 came from NOAA's 'Regional Bleaching Heat Stress Gauges' for the

#	Comment	Response
	35, is that correct? Is this reference using a 2 by 2 degrees box?	Society Islands. The area for this heat stress gauge is defined by NOAA's Society Archipelago Regional Virtual Station – based on personal communication (Erick Geiger) this station covers 1170 pixels of 5 km ² , which is equivalent to approximately 29,520 km ² (but includes land pixels). This is about 35 % smaller than a simple 2 x 2 degree box.
2.14	Fig.S1a: The blue and green lines are hard to distinguish, chose another color for one of them. Then, your lines are all hidden behind the red one. Maybe use different linewidths or something else to clearly distinguish these lines.	We used ColorBrewer to produce a palette of the 3 most distinct colour considered “colorblind safe” and are unaware of better colours to differentiate these. The important details for the specific events are shown in panels b and c.
2.15	L.135: add a reference to Fig.1a,b after (30.2degC).	Added.
2.16	L.143: Is p the correlation coefficient?	Apologies, this should have been r ² , and has been corrected.
2.17	L.145-146: This statement contradicts the previous sentence where the correlation was higher at 40m than 10m.	The order of the correlations has been corrected in the previous sentence.
2.18	L.147: Do you mean “attenuation” instead of “amelioration”?	Amelioration was intended but has been changed to attenuation as suggested.
2.19	L.180: “similar” is quite strong when comparing Fig.2p and Fig.2l.	We agree, the wording has been adjusted to remove the comparison (L192-193): since variability during 2019 (Fig. 2p) was around a warmer background temperature closer to the coral bleaching threshold.
2.20	L.194: replace “which” by “with”?	Corrected.
2.21	L.197: add “between” at the beginning of the line.	Added.
2.22	L.206: Should it be Fig.2g-h instead of Fig.2c-d?	Yes, thank you. Corrected.
2.23	L.237: Fig.5 seems to be mentioned before Fig.4.	Thank you, figures have been reordered.
2.24	L.256: Fig.4b should be Fig.4a.	Corrected (now Fig 5a).
2.25	L.258: Fig.4c should be Fig.4b for Mar-Apr period.	Corrected (now Fig 5b).
2.26	L.269: Replace “depths < 50 m” by “depths shallower than 50 m” and similarly for other occurrences. Inequality signs are confusing specifically when you're using positive	Inequality signs have been corrected here and replaced throughout the text.

#	Comment	Response
	depths in the text but negative depths in the figures (Fig.1,...).	
2.27	L.269: replace “surface mixed layer” by “mixed layer depth”, same l.273.	“surface” removed here (L296) and on what was L273 (L300)
2.28	L.274: “...thermocline is more...”.	Corrected.
2.29	L.316: Cut the sentence, the link is unclear between the 2 clauses.	Sentence corrected as follows: Coral mortality following bleaching is an important aspect of the ecological impact of extreme ocean temperatures, with mortality expected to be more widespread as thermal stress events caused by MHWs increase in frequency and spatial extent (L347-349).
2.30	L.580: The temperature recorders at 4 different depths are at different locations. This was unclear from only reading the text and figures. You could add a subplot in Fig.S1 by zooming north of Moorea and showing the locations of these recorders. Alternatively, you could add the latitude/longitude of each instrument in Table S1.	The wording has been updated to make it clear that the temperature recorders were bottom mounted directly onto the reef at each depth, with a horizontal distance in between: ...Seabird Electronics SBE39 and SBE56 temperature recorders (0.002 °C accuracy, 0.0001 °C resolution, less than 10 s response time) bottom-mounted directly onto the reef surface, i.e., they measured water temperatures immediately overlying the coral communities of interest. The horizontal distance between these loggers was about 40-100 m based on the reef slope at Moorea varying between 10 to 25 % (L632-636) As suggested a subplot has been added to Fig S1 in an attempt to better show the logger locations relative to the reef bathymetry (Figure S1e). Unfortunately, only the latitude/longitude of the shallow most logger is recorded (deeper loggers are located by divers swimming down slope from the shallow logger).
2.31	l.616: I don't understand the meaning of “to isolate removed frequencies”.	“Removed” was an error and has been deleted (L671).
2.32	L.625: You missed a word after also, maybe add “calculated”.	Thank you, “calculated” added (L700).
2.33	It's not clear from the paragraph l.613 why you would calculate IWC with eq.2 instead of simply applying a high-pass filter.	As detailed in our response to Reviewer 1 (#1.3-1.7), and described in Wyatt et al. (2020), a simple low-pass filter would not accurately represent the impact of internal waves interacting with the reef slope on reef-level temperatures, and thus conversely the high-pass filter would not represent the internal-wave cooling adequately. The effect of the internal waves at reef level are periodic drops in temperature as the waves break / run up the slope – they do not represent periodic oscillations about a mean state that could be reflected in a simple filtering approach.
2.34	L.631: Is n the number of photoquadrats, clarify and maybe change notation as n is already used l.596?	Quadrats specified and “n” removed (L706).

#	Comment	Response
2.35	L.642: How do you define “offshore from Moorea” exactly? Are you using a box around Moorea or one of the boxes in Fig.S1d? What is the depth associated with this box given that Argo floats are mostly in deep water? Specifying this should help to understand l.252 and l.816. For example, l.252 refers to deep water below 1500 m whereas Fig.4a stops at 200 m deep.	This text has been updated to make it clear that this is a box around Moorea: Regional hydrographic information in a 2 ° × 2 ° box around Moorea (see Fig. 1a-b) (L717-718) This box includes water depths up to 5000 m depth but such depths are largely irrelevant to the area of our interest, which is the upper 200 m as shown in Fig 5a. The text reference to 1500 m is simply to highlight the fact that deep ocean waters surrounding Moorea.
2.36	Table 1: Try to be consistent by reporting numbers with two decimals everywhere, see for example column 2 and 3.	We have reported to three significant figures throughout the table since we feel a decimal place limitation would be less useful for values that vary over orders of magnitude.
2.37	L.806: “shown for reference”.	It is unclear what issue is being raised here, but the caption has been adjusted to: “The satellite-derived sea surface temperatures are shown for comparison to in-situ temperatures (SST; grey line)” (L881-882)
2.38	Fig.5b: Use a different color than the dashed grey line, it’s hard to see behind the black curve.	We chose the dashed line in what is now Fig 4b to match the satellite SLA from what is now Fig 5b-c and the grey to indicate a satellite measure (grey is used as standard for SST, e.g., Fig 2). We can’t identify another colour that would not be confusing in the context of what is now Fig 5b-c.
2.39	L.834: b-c should be c-d.	Corrected, thank you.
2.40	L.894-896: Unclear and long sentence.	This sentence has been divided for clarity (L982-987).
2.41	Extended data Fig.2: I can’t really distinguish the black/blue lines/contours, maybe use another color. The fontsize of the figure is too small. Maybe do a plot in 2 by 2 subplots, also expand the x-axis width, this could probably help to remove the tiny insets with illegible captions.	We have adjusted this figure to make it a 2 x 2 subplot and increased the font size of the insets (L929). We trust it is now more legible. We used ‘ColorBrewer’ colour scales in the manuscript, which are designed to provide optimally distinct, “colorblind safe” colour palettes.
2.42	L.897: MHWs. “the most severe events”.	Corrected, thank you.
2.43	L.898: Fig.S1a-b is about DHD not SST.	Corrected, thank you.
2.44	L.900: I don’t understand where “17 days” comes from, Tab.S2 would say 14 days.	Typo corrected to 14.
2.45	L.906: Provide a reference to the Nino 3.4.	Reference to NOAA monthly ERSSTv5 added: No significant relationships were found between the Nino 3.4 Anomaly (monthly ERSSTv5 (NOAA, 2020)) during, or in the weeks to months before, the event

#	Comment	Response
		and the severity of the MHW as the maximum DHD. (L997-999)
2.46	L.906: Should the second comma be after “the event”?	We do not believe so, the ‘event’ is needed for both the ‘during’ and the ‘before’.
2.47	L.909: not 2 but 3 MHWs not evident at a regionals scale, what about April 1991?	Corrected.
2.48	L.910: (Fig.S1a; Table S3 vs Table S2).	Corrected.
2.49	L.911: “detection” instead of “localization”.	We have adjusted the text to make it clearer that this was explicitly focused on the localisation of the 2019 MHW over the Moorea’s north shore rather than just its detection: The localisation of the 2019 MHW over Moorea’s north shore was especially evident... (L1003)
2.50	L.939: Should it be Fig.S1a instead of Fig.S1b?	Corrected, thank you.
2.51	L.945: Replace “interrogate”.	We respectfully feel that this is the correct word in this instance.
2.52	L.947: Should it be Fig.S1a instead of Fig.S1b?	Corrected, thank you.
2.53	L.982: What is pseudo-F and p?	Text has been added to clarify pseudo-F: Coral cover data was analysed using permutational multivariate analysis of variance (PERMANOVA) within PRIMER v6 (PRIMER-E, Quest Research Limited, Auckland, NZ). PERMANOVA uses permutation of the data to construct an F statistic called ‘pseudo-F’(Anderson, 2001). (L1074-1076)

References (Responses to Reviewer #2)

- Anderson, M.J. (2001) A new method for non-parametric multivariate analysis of variance. *Austral Ecology* **26**: 32-46.
- Barott, K.L., Huffmyer, A.S., Davidson, J.M., Lenz, E.A., Matsuda, S.B., Hancock, J.R., Innis, T., Drury, C., Putnam, H.M. and Gates, R.D. (2021) Coral bleaching response is unaltered following acclimatization to reefs with distinct environmental conditions. *Proceedings of the National Academy of Sciences* **118**: e2025435118.
- Bates, A.E., Helmuth, B., Burrows, M.T., Duncan, M.I., Garrabou, J., Guy-Haim, T., Lima, F., Queiros, A.M., Seabra, R., Marsh, R., Belmaker, J., Bensoussan, N., Dong, Y., Mazaris, A.D., Smale, D., Wahl, M. and Rilov, G. (2018) Biologists ignore ocean weather at their peril. *Nature* **560**: 299-301.
- NOAA (2020) *Monthly Atmospheric & SST Indices*. Climate Prediction Service, National Weather Service, National Oceanic and Atmospheric Administration. (<https://www.cpc.ncep.noaa.gov/data/indices/> - 14 May 2020).
- Wyatt, A.S.J., Leichter, J.J., Toth, L.T., Miyajima, T., Aronson, R.B. and Nagata, T. (2020) Heat accumulation on coral reefs mitigated by internal waves. *Nature Geoscience* **13**: 28-34.

#	Comment	Response
3.1	Overall, I think that this is a compelling paper that emphasizes the need to understand the local and regional marine weather that can affect the vertical distribution of heat on coral reefs. It builds on our understanding of the role of internal ways in mitigating MHW, and introduces the role of the regional eddy field in either emphasizing or quelling the potential for IWC to mitigate MHWs.	Thank you for your positive comment on the compelling nature of the findings.
3.2	I am not aware of other data sets that include such a long history of in situ temperature records, although I suspect that they should be available from the GBR through the efforts of AIMS. But I have not seen an analysis of the role of eddies in this context. The length of the record and the associated open ocean profile data provide a unique perspective these processes.	The reviewer is correct that, unfortunately, the time-series records for this kind of analysis are sorely lacking. This has limited our capacity to understand the dynamics of reef thermal environments and thus explain patterns in, and make predictions regarding, coral bleaching responses. We are unaware of long-term data with sufficient temporal and reef-depth resolution (along with corresponding oceanographic measurements) from the GBR, but analysis of internal wave influences on thermal dynamics of the GBR would be a very important analysis if this data is available. As below (#3.4), the inshore GBR might be one of the few locations where internal wave dynamics are more limited, but we expect the majority of ocean-exposed and outer reefs to experience thermal variability in response to internal waves.
3.3	One note that I would make is that ICW may also contribute the flux of a planktonic food supply. Unfortunately, the measurements for this are harder to come by, although more recent technology may facilitate those types of measurements. The damping of the ICW may damp the supply of a planktonic food supply to the corals which could reduce the ability of the coral to adapt to the effects of the MHW.	We agree that the nutrient / food supply associated with internal waves is an extremely interesting topic. While it is part of our ongoing (but sadly COVID-19 disrupted) research, we felt it was currently too speculative to emphasise in the paper. As it is of ongoing interest and raised by the reviewer, we have added some text to highlight this important avenue for ongoing research: In addition to cooling effects, future research could examine the potential role of internal waves in bleaching mitigation through upwelling of organic particles associated with the deep chlorophyll maximum (Leichter et al., 1996; Leichter et al., 1998; Leichter & Genovese, 2006; Wyatt et al., 2020), which could enhance coral feeding and also improve outcomes for thermally stressed corals (Grottoli et al., 2006; Rodrigues & Grottoli, 2007; Huffmyer et al., 2021). (L400-403)
3.4	1. Line 219 – perhaps the majority of reefs do experience “some degree of IWC”. But there are regions where IWC is a rare occurrence.	We agree that some, but not all, inshore reefs might experience limited internal wave exposure but feel that many studies are too quick to dismiss the presence of internal waves, sometimes based on low-frequency data that is incapable of capturing such dynamics. Internal-wave cooling should be considered a potentially influential process in ocean-exposed coral’s thermal environment unless high resolution data demonstrates an absence of internal waves.
3.5	2. Line 224 – Some of the “processes seaward of reef slopes” may themselves bring cooling to the reef. Anticyclonic eddies have uplifted isopycnals along their periphery. Depending on the water column structure these in themselves may alter the temperature environment on the reef. This is opposite in effect to what is described in this paper. It’s not contradictory, but rather depends on how the oceanographic feature	We agree that this is a process of complex interactions between an eddy field and reef slope. It is conceivable that an anticyclonic eddy, if the isopycnals were uplifted at its margin, could increase reef cooling if only the margin of the eddy interacted with the reef slope (and any concomitant impact on internal-wave dynamics were ignored). This would be analogous to the effect of the cyclonic eddy in terms of isopycnal uplift, but the internal wave dynamics in this specific case are unclear. Our data suggest that any cooling during the passage of anticyclonic eddies is likely to be highly stochastic given the edge effects would exist for a very short

#	Comment	Response
	intersects with the shallow structure of the reef.	relative time period. Given the short period of this possible process, it appears to be negligible relative to the major impact these eddies have on depressing isopycnals, and thus the depth of internal waves and reef-level temperatures, over a large spatial area for days to weeks at a time. Even at the eddy periphery, ARGO casts in 2019 suggest reduced stratification and mixed layer deepening which would lead to reduced internal-wave cooling on reefs (see new Extended Data Figure 4)
3.6	3. Line 283 – the “SLAs” are propagating. Should we deduce that the eddies are westward propagating as well?	The SLAs are a manifestation of the eddies, so we expect that the westward bands of SLA show eddies propagating westward. This can be seen in the Supplementary Movie files 3 and 4. The westward propagation of SLAs (eddies) past Moorea can be seen especially clearly in the latter.
3.7	4. Line 344 “centred” – Is American or UK English being used?	We have attempted to use English (UK) as standard throughout.
3.8	5. Line 352 – when “nutrient upwelling” is mentioned, is that inorganic nutrients, or planktonic food supply? I suspect with the mention of bleaching that it refers to inorganic nutrients. An interesting question is whether with the suppression of internal waves, planktonic nutrition to the reef is also suppressed, perhaps exacerbating the effects of the MHW.	The reviewer raises a very interesting point. Here we were specifically referring to inorganic nutrient upwelling since some recent work has focused on linking coral bleaching to increased inorganic nutrient loading. The clear link between internal waves and bleaching in this data puts the focus firmly back on temperature as the main driver, but the question of whether internal waves increase particulate organic matter (plankton) supply, and importantly ingestion by corals, thus leading to an additional protective effect of internal waves is certainly a focus for future research that the present findings might should motivate (see also #3.3).

References (Responses to Reviewer #3)

- Grottoli, A.G., Rodrigues, L.J. and Palardy, J.E. (2006) Heterotrophic plasticity and resilience in bleached corals. *Nature* **440**: 1186-1189.
- Huffmyer, A.S., Johnson, C.J., Epps, A.M., Lemus, J.D. and Gates, R.D. (2021) Feeding and thermal conditioning enhance coral temperature tolerance in juvenile *Pocillopora acuta*. *Royal Society Open Science* **8**: 210644.
- Leichter, J.J. and Genovese, S.J. (2006) Intermittent upwelling and subsidized growth of the scleractinian coral *Madracis mirabilis* on the deep fore-reef slope of Discovery Bay, Jamaica. *Marine Ecology-Progress Series* **316**: 95-103.
- Leichter, J.J., Shellenbarger, G., Genovese, S.J. and Wing, S.R. (1998) Breaking internal waves on a Florida (USA) coral reef: a plankton pump at work? *Marine Ecology-Progress Series* **166**: 83-97.
- Leichter, J.J., Wing, S.R., Miller, S.L. and Denny, M.W. (1996) Pulsed delivery of subthermocline water to Conch Reef (Florida Keys) by internal tidal bores. *Limnology & Oceanography* **41**: 1490-1501.
- Rodrigues, L.J. and Grottoli, A.G. (2007) Energy reserves and metabolism as indicators of coral recovery from bleaching. *Limnology & Oceanography* **52**: 1874-1882.
- Wyatt, A.S.J., Leichter, J.J., Toth, L.T., Miyajima, T., Aronson, R.B. and Nagata, T. (2020) Heat accumulation on coral reefs mitigated by internal waves. *Nature Geoscience* **13**: 28-34.

Reviewer #1 (Remarks to the Author):

Reviewer #1 Attachment on the following page

Review of manuscript : Hidden heatwaves and severe coral bleaching linked to mesoscale eddies and thermocline dynamics

Wyatt et al., submitted to Nature Communications

September 8, 2022

This is my second review of the manuscript. The authors replied to my comments point by point, I thank them for that. Overall, I am satisfied with their responses to comments # 1.8 to # 1.19 (according to their labelling). However, I had made an important comment on the filtering approach and the classification of wave vs non-wave signals (a comment that has been split into comments # 1.4 to # 1.7). This methodological point is key in their analysis and likely has a significant impact on their results. The authors have not convinced me that their methodology is adequate and scientifically correct. I cannot recommend the manuscript for publication in the present form.

1 Major comments

The authors' quotes from the rebuttal are italicized.

1.5 *the goal was not to achieve a standard frequency filtering outcome, but rather to approximate what the low-pass component of the temperature signal would have been had there not been imbedded high-frequency downward temperature deviations.*

The downward temperature deviations are immediately followed by upward temperature deviations, also associated with internal waves – see, e.g., their Figure 2l. Consequently, there is a deliberate bias in choosing to consider only the downward temperature deviations.

1.5 The authors base their rebuttal on *visual and empirical analysis [...]*, *Again, visual and empirical analysis [...]*, *visual examination [...]*, which belong to the lexical field of human sensations and cannot be used to elaborate scientific results. If visual inspection hints that there are more cooling events than heating events by internal waves, then this should be backed up by a diagnostic, e.g., a distribution of temperature deviation. Then, if the distribution is actually not symmetric, the authors could design a criterion to identify cooling events. Figure 1 here is a copy of Figure 2l in the manuscript, where I added an alternative interpretation of the non-wave signal (green line) to emphasize the subjectivity of the criterion.

1.5 *visual and empirical analysis that the downward deviations in temperature over these shallow reef habitats (above the mean thermocline depth) are well approximated by the RMS variability within the semi-diurnal frequency band: $RMS(HP)$*

Why $RMS(HP)$ and not $0.5 \times RMS(HP)$? or a given percentile of HP ?

1.5 *the internal wave filtering approach described in Wyatt et al. (2020) is not central to the main finding of the present paper [...]* We believe this main finding would stand regardless of the

Figure 1: A copy of original Figure 21. The green line is my visual interpretation of non-internal-wave temperature variation, to be compared to the original white line.

filtering approach taken to estimate a non-internal-wave signal.

The filtering approach is key in the identification of wave vs non-wave signals. Most of the results presented in the manuscript depend quantitatively and qualitatively on the method.

1.6 *application of the method we describe to the more general case of thermistor records on a vertical mooring is a different situation than bottom-mounted thermistor time series in shallow water.*

I agree that the dynamics are different in the open ocean and above a shallow-water reef, and methodologies to identify wave vs non-wave signals could differ. However, I think the authors must carefully design their methods based on diagnostics on the raw data, not only on visual inspection.

Reviewer #2 (Remarks to the Author):

I found that the authors addressed most of my previous concerns as well as the ones raised by the other Reviewers. I would advise to accept the manuscript.

I still found several minor corrections described below:

I.1146: ‘...low pass (LP; blue dashed line) filtered with a running mean filter to remove variance within the semi-diurnal band (i.e., the local inertial frequency at Moorea, 39.9 hrs).’

This is unclear, it would be clearer to say ‘... with a running mean filter over an inertial period of 39.9 hrs at Moorea to remove variance within the semi-diurnal band.’

However, this also seems to contradict your definition I.671 where you only refer to a low and high-pass filter around the inertial period. You don’t refer to a running mean in this section or even to the semi-diurnal band, which is referred in the main text. Explain and clarify these contradictions between the Methods section and Fig.S2.

I.695: I think your explanation remains partly confusing. An easy improvement would be to add the third equation that reviewer #1 sent in his first round of review (third equation below his Fig.1).

I.1149: Why is it 39.3 hrs and 39.9 hrs 3 lines above? Similarly, I.672 and I.688.

You haven’t solved the issue of decimals in Tab.1

For example, you have in column 5:

5.73 and 0.0831 and it should be 5.73 and 0.08.

#	Comment	Response
1.1	This is my second review of the manuscript. The authors replied to my comments point by point, I thank them for that. Overall, I am satisfied with their responses to comments # 1.8 to # 1.19 (according to their labelling).	Thank you for re-reviewing our manuscript and spending considerable time providing detailed comments which have helped us to further clarify and improve it. We are pleased that our extensive revisions, including new data and analysis, have satisfactorily addressed the major points of the prior reviews. We are also pleased to note that the revised manuscript is considered ready for publication by the other reviewers.
1.2	However, I had made an important comment on the filtering approach and the classification of wave vs non-wave signals (a comment that has been split into comments # 1.4 to # 1.7). This methodological point is key in their analysis and likely has a significant impact on their results. The authors have not convinced me that their methodology is adequate and scientifically correct. I cannot recommend the manuscript for publication in the present form.	We believe we did respond in great detail to the reviewer's comments, in the previous revisions and response comments, on the use of the published filtering approach to empirically estimate an internal-wave-free time series for comparing cooling between marine heatwave (MHW) events. To clarify further, the purpose of the filtering approach is to provide a consistent, quantitative means of comparing variability among MHW events across the depth-specific time series at our study site. While the main findings of the current manuscript do include comparisons among events, specifically with respect to estimated internal wave cooling (IWC), this is only one point of the manuscript and arguably not key in the overall findings. Below we show that the specific methods of those comparisons do not alter the key findings and that the important differences among events are evident in the raw data in addition to the analysis via filtering. Also, we stress that the key points of the manuscript include the importance of spatial and temporal scales in assessing MHWs, insight into potential causes of the differences among events driven by mesoscale oceanographic eddies, and the corresponding differences in coral bleaching severity during recent MHWs. These key findings of the manuscript are the result of analyses of the raw reef-level temperature data, coupled with ancillary time series observations of water column stratification, sea-level anomalies, and ecological monitoring. None of these data are sensitive to the filtering methodologies applied to the reef-level temperature time series. The observed differences in temperatures variance and estimated internal wave cooling between events support the larger key findings, but again are not critical to those findings. Below we also attempt to demonstrate further that our filtering approach is robust and appropriate for the specific application of estimating the background temperature patterns within the reef-level temperature time series that would have been observed had the repeated cooling events produced by the run up of internal waves not been present. While we agree there could be other ways of estimating an 'internal-wave-free time series', specific advantages of the approach here include that the method is well-characterized, previously peer-reviewed and published, and straight-forward to understand – being just the simple application of a low-pass filter and adjustment of the resulting signal based on the root-mean-square of the high-frequency variability that results from repeated bouts of cooling produced by the run-up of internal waves on the reef slope. Below (Figure A) and in the new Fig S2 now we provide further quantitative analysis showing that the high-frequency thermal variability in our time series results predominately from specific events of (1) rapid downward temperature deviations as cool, subsurface water is advected upward onto the reef followed closely by (2) warming associated with a return to pre-event, ambient conditions. We also note that multiple qualified reviewers of the methodology published in Nature Geoscience were satisfied with the simple filtering approach, and the other reviewers of the present manuscript appear satisfied. We understand that the reviewer may have interests (which we share) in the detailed physics behind internal waves interactions with reef slope topography. However, we believe that complex topic is outside the scope of the present contribution. Here we are examining the implications of mesoscale oceanographic eddies on the dynamics of water column stratification and reef-level thermal regimes, and associated coral bleaching events. Our study was primarily motivated by, and

#	Comment	Response
		the analyses explain, the surprising differences in the observed coral bleaching among recent MHWs that appear paradoxical based on sea surface temperatures (SST). The detailed and complex physics of internal-wave breaking and interactions with a reef slope would be a different study entirely. As we pointed out previously, and reiterate below, the manuscript's major findings are not sensitive to the data filtering approach. Again, we believe the approach does provide a simple, quantitative means of estimating cooling due to internal waves and for consistently comparing across heatwave events in multiple years. However, the majority of our analyses are independent of any filtering (see #1.6) and support the same overall conclusion of differences between heatwaves.
1.3	The downward temperature deviations are immediately followed by upward temperature deviations, also associated with internal waves – see, e.g., their Figure 2l. Consequently, there is a deliberate bias in choosing to consider only the downward temperature deviations.	This comment may represent a misunderstanding of our field setting or our descriptions of the dynamic temperatures observed at the study site. In any case, there is no deliberate bias or choice to only consider downward temperature deviations in our analyses. If temperatures did not raise again, internal-wave cooling would be permanent. This does not represent the physical case of cool water being advected onto the reef and subsequently receding off the reef slope, previously documented at our study site and multiple other coral reef locations around the globe. Rather, temperature drops associated with the arrival of internal waves on the reef slope subsequently increase as the waves recede. Analysis of the raw temperature data shows that the “upward temperature deviations” the reviewer refers to represent temperatures returning to pre-event conditions as cool water advected onto the reef by internal waves subsequently recedes. In response to the reviewer request we have also conducted a new quantitative analysis of the timing of cooling and subsequent warming events. We show this return to pre-event temperatures between internal waves in what is now Fig. S3 (see Figure B) and have added further analysis (see #1.4, Figure A). We are specifically focused on the cooling influence of internal waves on the reef-slope community, and thus in the “downward temperature deviations” shown clearly in Fig. 2 and what is now Fig. S3 (again, see Figure B). As we explained previously, it is impossible to precisely determine a mean background temperature in the absence of internal waves, but we can make a best estimate that reflects what we know about how non-linear internal waves interact with reef slopes at our study site and elsewhere. Our simple approach to this estimate is robust and suitable for the purpose at hand. We also note below that any ‘warm bias’ associated with adding the rms(HP) to the low-pass signal would be in the direction of decreasing the differences between MHWs in 2016 and 2019, which would reduce rather than amplify the comparisons that are central findings of our current manuscript.
1.4	The authors base their rebuttal on visual and empirical analysis [. . .], Again, visual and empirical analysis [. . .], visual examination [. . .], which belong to the lexical field of human sensations and cannot be used to elaborate scientific results. If visual inspection hints that there are more cooling events than heating events by internal waves, then this should be backed up by a diagnostic, e.g., a distribution of temperature deviation. Then, if the	We fear the reviewer may misunderstand aspects of our responses and the crux of our explanation may not have come through as clearly as we would have liked – principally that the internal-wave filtering provides an estimation for comparison between heatwave events, and that the specific filtering methods are not integral to the manuscript's overall findings. We mentioned visual inspection of the data to encourage the reviewer to consider the data, and the context for internal-wave interactions with reef slopes, rather than representing internal-wave signals from the open ocean as relevant to the question at hand. In considering the data, we direct the reviewer to the observations (black lines) of Figures 2a-d and Figures 2e-h. These clearly show predominately negative temperature anomalies due to cool water advecting up the reef slope due to non-linear internal waves. The waves cause lower temperature pulses that are superimposed on slowly varying, warmer background temperatures. The magnitudes of the cooler pulses increase with depth which is consistent with cool water intermittently coming up from deeper depths.

#	Comment	Response
	distribution is actually not symmetric, the authors could design a criterion to identify cooling events. Figure 1 here is a copy of Figure 2I in the manuscript, where I added an alternative interpretation of the non-wave signal (green line) to emphasize the subjectivity of the criterion.	Regarding the request to “design a criterion to identify cooling events” we have now added an analysis and supplementary figure (new Figure S2, reproduced as Figure A below) in which we identify major temperature variance events and confirm the non-linear nature of the internal-wave events on the reef slope: cooling is followed by warming back to pre-event conditions, with cooling occurring about 2 hrs before a similar magnitude of warming. Description of this analysis has been added to the manuscript (L692-696). While we hope this analysis and graphical representation is helpful, we note again that the differences in the prevalence of internal waves between heatwaves is clear in the observed data and other analyses independent of the estimation of internal-wave cooling by filtering.  Figure A: Visualisation of the approach used to identify the non-linear nature of major internal wave events in in-situ temperature time series observed over the reef slope. The prevalence of cooling followed by warming back up was identified by taking (a) the water temperatures observed (black line) at reef-level (the example here is for the 20 m isobath on Moorea’s north shore reef slope during 31 March to 10 April) and identifying the (b) temperature differential (averaged over a two-hour period to give ΔT_{avg}). (c) Major events were arbitrarily defined as ΔT_{avg} above or below $0.015\text{ }^{\circ}\text{C}$, with negative (cooling, blue) and positive (heating, red) events occurring within 10 hrs of each other identified with arrows. Cooling occurred 2.6 ± 1.3 hours before warming since 2016 (compared to 2.2 ± 0.6, 2.2 ± 1.3, at 2.0 ± 2.0 at 10, 30 and 40 m). Multiple events within close proximity (e.g., 31 Mar 2016) were excluded from the temporal analysis but showed the same trend of cooling followed by warming. [new Fig S2 in manuscript] It’s not clear why the reviewer recreated a version of Figure 2I (which is not focused on the filtered data) when we provided a detailed figure in the revision (now Fig S3; reproduced as Figure B below) in response to the previous comments – the reviewer did not mention this requested explanation. It clearly demonstrated how the NIW signal closely matches the (warm) temperatures prevailing between internal waves when they are present. Perhaps the reviewer missed the addition of what is now Fig. S3 and its support of our approach to estimate the temperature time series if internal-wave cooling was absent and to compare patterns of temperature and temperature variability between MHW events.

#	Comment	Response
		 Figure B: Reproduction of Fig. S3 from the manuscript for reference. Visualisation of the filtering approach described in Wyatt et al. (Wyatt et al., 2020) used to estimate the influence of internal waves on the observed in-situ temperature time series. Panels show water temperatures observed (black line) at reef-level on the 20 m isobath on Moorea’s north shore reef slope during 27 March to 3 April in (a) 2016 and (b) 2019, when internal waves were prevalent and absent, respectively. The observed temperatures were low pass (LP; blue dashed line) filtered over an inertial period of 40.0 hrs at Moorea to remove variance within the semi-diurnal band. The non-internal-wave (NIW; red line) signal was then derived by shifting the LP signal upwards by an amount corresponding to the root mean square (again, within a 40.0 hr window) of the high frequency variability to reflect the observation that the influence of internal waves on temperatures observed at reef-level above the typical thermocline depth is predominately in the form of rapid, downward spikes in temperatures as the internal waves push isotherms from below the surface mixed layer up onto the reef slope (see a). The running mean LP signal includes the net cooling caused by internal waves and therefore tends to (a) underestimate prevailing temperatures in the absence of internal waves when internal waves are present in the observed signal, but (b) closely resembles the NIW and observed signals when internal waves are largely absent. [now Fig S3 in manuscript] We do also note that the raw data from 40 m depth collected during 2016 and shown in Figure 2I were incorrectly sampled at a 2-hour instead of 2-minute interval. Those data which the reviewer focussed on in providing an “alternative interpretation” do not show the data at sufficient resolution to identify the non-linear nature of internal wave cooling events. We now note the data collection error in the manuscript, L642-644 and L920-921) and figure caption.
1.5	Why RMS(HP) and not 0.5×RMS(HP) ? or a given percentile of HP ?	We agree, as stated in great detail here and in previous responses, that it is difficult to objectively determine a “non-internal-wave” time signal from one that contains these waves. However, as stated previously, and in the original peer-reviewed paper presenting the filtering approach (Wyatt et al., 2020), the reason for adding the rms(HP) to the filtered low-pass temperature signal is that the high-frequency temperature variability predominantly results from rapid cooling events followed by warming back to pre-event conditions, rather than symmetrical oscillations around a mean state. In developing and testing the filtering approach on real and simulated time series we find that adding the rms(HP) to the low-pass signal well approximates the background temperatures absent the cooling events. We again emphasise that we have used a consistent and simple approach based on a previously published study (Wyatt et al., 2020). We feel the consistent application we have described is a reasonable approximation to a possible ‘non-internal-wave’ signal. We stress again though that the filtering approach does not alter the overall findings of the manuscript which are almost entirely based on observed data (e.g., see Figure D).

#	Comment	Response
		As already explained, the rms(HP) adjustment was introduced to remove the cooling from internal waves evident in the raw data, as can clearly be seen in what is now Fig. S3 (reproduced in Figure B). To even more clearly demonstrate this point, we created a synthetic time series that represents a static temperature combined with semi diurnal cooling, showing that the NIW signal exactly reproduces the static temperature, while the LP contains an unwanted cold bias (Figure C). We earlier attempted to make this clear in the text in response to the reviewer's previous comments: "the low-pass signal contains the effects of net cooling from internal waves interacting with the reef slope" (L704-705). $0.5 \times \text{rms(HP)}$ would also underestimate the cooling associated with non-linear internal waves.  Figure C: Demonstration of the internal wave filtering approach on synthetic temperature data. (a) A static temperature timeseries (29.5 °C) has (b) semi-diurnal cooling of up to 1.5 °C applied to it to produce (c) the simulated observed (Obs.) timeseries (black line). The non-internal-wave (NIW) filtered data (red line) exactly reproduces the static temperature while the low-pass (LP) filtered data (blue dotted line) has an unwanted cold bias associated with superimposed cooling events.
1.6	The filtering approach is key in the identification of wave vs non-wave signals. Most of the results presented in the manuscript depend quantitatively and qualitatively on the method.	Again, as explained above, the filtering approach is in fact not key to the overall findings of the manuscript. The filtering is used to highlight internal-wave cooling differences between events (in just one of 12 figures and extended data figures) and is supportive, but not integral, to the overall findings. We now further show that we can equally compare the temperature variability between events in a simple quantitative fashion as the daily temperature variance based on the observed, unfiltered temperature data (Figure D). The overall findings regarding the direction and magnitude of differences among MHWs across years remain the same. We now include a panel showing the daily temperature variance along with the IWC in the figure (Fig. 3b and Fig 3a, respectively), and reproduce those panels here for clarity.

#	Comment	Response
		Figure D: Comparison of two methods for comparing internal-wave influences on reef-level temperatures during marine heatwave events: (a) average internal-wave cooling (IWC, in °C), as used in the manuscript based on a published filtering approach to specifically isolate cooling effect of internal waves (now Fig. 3b), or (b) daily temperature variance (in °C²) which captures internal-wave-induced and other variance at a coarser daily resolution (now Fig 3a).
1.7	I agree that the dynamics are different in the open ocean and above a shallow-water reef, and methodologies to identify wave vs non-wave signals could differ. However, I think the authors must carefully design their methods based on diagnostics on the raw data, not only on visual inspection.	We do not use visual inspection for diagnostics. In fact, in designing the filtering approach presented in Wyatt et al. (2020) we relied on multiple quantitative diagnostics, including ensuring the estimated non-internal-wave signal did not exceed maximum daily temperatures actually observed between internal wave cooling events, and showing that the non-internal-wave signal very closely reproduces the observed signal during seasons and time periods when internal wave impacts were absent (e.g., winter and after storm mixing). As detailed in #1.4 our suggestion for visual inspection of the data was offered to highlight the non-linear nature of internal waves impacting the reef slope. This is now supported by diagnostic patterns on the raw data evident from the rates of negative and positive temperature changes shown in the new Fig S2 (Figure A). The cooling and warming are not uniformly distributed in the time series (as would be the case for purely symmetrical oscillations around a mean temperature); rather re-warming consistently closely follows cooling events. Our approach for estimating the cooling effect of internal-waves on reef-level temperatures, and for comparing overall cooling between marine heatwave events, is quantitative, being based on calculated quantities within the data (low-pass data plus rms(HP)), and backed up by direct observations of the run-up of internal waves on sloping topography described in prior literature. We offered the comment on visual inspection of the data simply to reinforced the details explained in the previous responses, as well as the revised manuscript and published in Nature Geoscience (Wyatt et al., 2020).

References (Responses to Reviewer #1)

Wyatt, A.S.J., Leichter, J.J., Toth, L.T., Miyajima, T., Aronson, R.B. and Nagata, T. (2020) Heat accumulation on coral reefs mitigated by internal waves. *Nature Geoscience* **13**: 28-34.

#	Comment	Response
2.1	I found that the authors addressed most of my previous concerns as well as the ones raised by the other Reviewers. I would advise to accept the manuscript.	Thank you for your positive comments that have helped improve the manuscript and ensured it is ready for publication.
2.2	I.1146: ‘...low pass (LP; blue dashed line) filtered with a running mean filter to remove variance within the semi-diurnal band (i.e., the local inertial frequency at Moorea, 39.9 hrs).’ This is unclear, it would be clearer to say ‘... with a running mean filter over an inertial period of 39.9 hrs at Moorea to remove variance within the semi-diurnal band.’ However, this also seems to contradict your definition I.671 where you only refer to a low and high-pass filter around the inertial period. You don’t refer to a running mean in this section or even to the semi-diurnal band, which is referred in the main text. Explain and clarify these contradictions between the Methods section and Fig.S2.	Apologies for the confusion. Mention of a ‘running mean’ filter in the caption was confusing. We have updated the caption text along the lines of the suggested changes, and it is now consistent with the description in the Methods: “The observed temperatures were low pass (LP; blue dashed line) filtered over an inertial period of 40.0 hrs at Moorea to remove variance within the semi-diurnal band.” (L1194-1195)
2.3	I.695: I think your explanation remains partly confusing. An easy improvement would be to add the third equation that reviewer #1 sent in his first round of review (third equation below his Fig.1).	We have clarified the derivation of the NIW as suggested: “To remove this net cooling effect from the low-pass signal, we estimated the root-mean-squared of the high-pass signal, rms(HP), within a moving window the size of the local inertial period (40.0 hrs) and added this quantity to the low-pass signal to estimate the NIW signal; i.e., NIW = LP + rms(HP)” (L707-710)
2.4	I.1149: Why is it 39.3 hrs and 39.9 hrs 3 lines above? Similarly, I.672 and I.688.	Thank you. These were typographical and rounding errors. We used the local inertial period of 39.966, which has been corrected to 40.0 hrs throughout the manuscript.
2.5	You haven’t solved the issue of decimals in Tab.1 For example, you have in column 5: 5.73 and 0.0831 and it should be 5.73 and 0.08.	We now have revised Table 1 to present two decimal places across all estimates. Previously, as a matter of style and accuracy, we had presented the data to 3 significant figures throughout, not decimal places. Both 5.73 and 0.0831 were presented to 3 significant figures.